# Transgenic expression of a T cell epitope in *Strongyloides ratti* reveals that helminth-specific CD4+ T cells constitute both Th2 and Treg populations

**Bonnie Douglas**[1], **Yun Wei**[2], **Xinshe Li**[1], **Annabel Ferguson**[1], **Li-Yin Hung**[1], **Christopher Pastore**[1], **Jonathan R Kurtz**[3], **James B. McLachlan**[4], **Thomas J. Nolan**[1], **James Lok**[1], **De'Broski R. Herbert**[1] *

1 Department of Pathobiology, University of Pennsylvania School of Veterinary Medicine, Philadelphia, Pennsylvania, United States of America, 2 Department of Oncology and Inflammation, Amgen Research, South San Francisco, California, United States of America, 3 Flagship Labs 72, Inc., Cambridge, Massachusetts, United States of America, 4 Department of Microbiology and Immunology, Tulane University School of Medicine, New Orleans, Louisiana, United States of America

* debroski@vet.upenn.edu

**Data Availability Statement:** RNA sequencing data set is publicly available on the NIH Gene Expression Omnibus (GEO) database. URL: https://

## Abstract

Helminths are distinct from microbial pathogens in both size and complexity, and are the likely evolutionary driving force for type 2 immunity. CD4+ helper T cells can both coordinate worm clearance and prevent immunopathology, but issues of T cell antigen specificity in the context of helminth-induced Th2 and T regulatory cell (Treg) responses have not been addressed. Herein, we generated a novel transgenic line of the gastrointestinal nematode *Strongyloides ratti* expressing the immunodominant CD4+ T cell epitope 2W1S as a fusion protein with green fluorescent protein (GFP) and FLAG peptide in order to track and study helminth-specific CD4+ T cells. C57BL/6 mice infected with this stable transgenic line (termed *Hulk*) underwent a dose-dependent expansion of activated CD44^hiCD11a^hi 2W1S-specific CD4+ T cells, preferentially in the lung parenchyma. Transcriptional profiling of 2W1S-specific CD4+ T cells isolated from mice infected with either *Hulk* or the enteric bacterial pathogen *Salmonella* expressing 2W1S revealed that pathogen context exerted a dominant influence over CD4+ T cell phenotype. Interestingly, *Hulk*-elicited 2W1S-specific CD4+ T cells exhibited both Th2 and Treg phenotypes and expressed high levels of the EGFR ligand amphiregulin, which differed greatly from the phenotype of 2W1S-specific CD4+ T cells elicited by 2W1S-expressing *Salmonella*. While immunization with 2W1S peptide did not enhance clearance of *Hulk* infection, immunization did increase total amphiregulin production as well as the number of amphiregulin-expressing CD3+ cells in the lung following *Hulk* infection. Altogether, this new model system elucidates effector as well as immunosuppressive and wound reparative roles of helminth-specific CD4+ T cells. This report establishes a new resource for studying the nature and function of helminth-specific T cells.

www-ncbi-nlm-nih-gov.ezproxy.u-pec.fr/geo/query/acc.cgi?acc=GSE134278 GEO accession: GSE134278 BioProject: PRJNA554640 SRA: SRP214620.

**Funding:** DRH was funded by NIH- U01 AI125940 and R21 A1144572 and by Burroughs Wellcome Fund under grant number 1013573.01- JBM was supported by NIH grant R01-AI-103343. The funders had no role in study design, data collection and analysis, decision to publish, or preparation of the manuscript.

**Competing interests:** The authors have declared that no competing interests exist.

## Author summary

Intestinal parasitic helminths infect roughly one billion people worldwide, and there are currently no vaccines available for use in humans. In humans and experimental mouse infection models, CD4+ helper T cells that have differentiated into type 2 (Th2) effectors serve important roles in worm clearance and are considered essential for specific, long-lasting immunity. However, many helminth infections also drive expansion of regulatory T cells (Tregs) that can suppress inflammatory CD4+ T cell subsets. Whether Th2 and/or Treg subsets recognize helminth antigens is a question of great relevance to vaccine development, but no tools previously existed to identify and study endogenous helminth-specific CD4+ T cells. Here, we used transgenesis in the *Strongyloides ratti* model to engineer the first gastrointestinal (GI) nematode strain to express a tractable CD4+ T cell peptide epitope, 2W1S (*Hulk*). Our studies reveal that 2W1S-specific CD4+ T cells become both Th2s and Tregs in the lungs of infected mice and potentially serve protective and/or suppressive roles during *Hulk* infection. Development of this new model organism could be an important tool for studies designed to understand Th2 and Treg immunobiology, microenvironment-specific interactions, helminth-epitope processing/presentation, and T cell-dependent antibody responses.

## Introduction

Parasitic helminth infections, including infections with soil-transmitted gastrointestinal (GI) nematodes, persist chronically and affect billions of individuals across the globe [1]. Anthelmintic drugs are effective at eliminating GI infections, but protective immunity often fails to develop against re-infection, emphasizing the need for vaccines to prevent these infections [2]. Though vaccine development has thus far proven unsuccessful [3], such efforts have significantly advanced knowledge of type 2 immune responses and highlighted the requirement of CD4+ T cells in controlling and clearing GI nematode infections [4–6]. However, the nature and function of CD4+ T cells specific for helminth-derived antigens remains largely unexplored.

Unlike microbial pathogens, which are relatively small in terms of both size and genomic complexity, parasitic nematodes are metazoan organisms that are orders of magnitude larger than most microbes and express a more diverse antigenic repertoire across multiple life stages within a single host [7–9]. Moreover, GI nematode infection induces considerable host tissue damage at sites of infection, including the respiratory and GI tracts. Tissue damage leads to release of IL-25, IL-33 and TSLP which, among other functions, can activate dendritic cells to present antigen to and activate CD4+ T cells towards the type 2 subset [10,11]. Activated type 2 CD4+ T cells (Th2) then migrate to affected tissues, where they both secrete type 2 cytokines, such as IL-4, IL-5 and IL-13, and also support ILC2 production of these cytokines in a contact-dependent manner [12–14]. Signaling through IL-4Rα and IL-5R can recruit and activate eosinophils, basophils, and mast cells to produce pharmacologically active compounds that damage worms, and IL-4 and IL-13 expand alternatively-activated macrophages (AAM) which can trap and sometimes kill larvae [12,15–17]. In tandem with IL-4Rα-dependent goblet cell hyperplasia and smooth muscle hypercontractility, these mechanisms drive the "weep and sweep" response that expels worms from the intestinal tract [10,18]. Overall, this complexity suggests that the corresponding CD4+ T cell response must be phenotypically diverse to coordinate clearance of GI nematodes, but whether Th2 cells need to be specific for helminth-derived antigens in order to drive these protective responses remains unknown.

In addition to coordinating pathogen clearance, CD4+ T cells also play important roles in preventing immunopathology and in tissue repair [19–26]. IL-4Rα-expressing Tregs are

essential for repairing tissue and restoring lung function in mice during infection with the GI nematode *Nippostrongylus brasiliensis* [23]. Further, when Foxp3-expressing cells are depleted prior to infection with *Heligmosomoides polygyrus*, host animals lose more weight and have worsened pathology associated with high levels of IFN-γ production [21]. However, when Tregs are expanded by administration of IL-2 complex, *H. polygyrus* expulsion is significantly delayed [21], and when Tregs are specifically deleted during *S. ratti* or *Litosomoides sigmodontis* infection worm burden and fecundity are reduced [22,27]. Altogether, this implicates Tregs as being essential not only for host tissue preservation but also for fine-tuning cytokine secretion and regulating worm burden. Whether the Treg compartment contains helminth-specific CD4+ T cells is largely unexplored due to a lack of tools to identify and track these cells during infection.

While bacterial and protozoan pathogens have been engineered to express model antigens (e.g. ovalbumin, 2W1S) from transgenes [28–30], most GI nematodes are currently refractory to stable, heritable transgenesis. However, *Strongyloides spp*. have proven uniquely amenable to transgenesis because this species can reproduce both asexually through parthenogenesis as parasitic adult females, or sexually as free-living, dioecious adults [31,32]. These features are important for two reasons: 1) the free-living (FL) adult females are viable outside the host and can be microinjected with plasmid-encoded transgenes and 2) the progeny of a given parasitic female are clonal, yielding higher frequencies of genetically identical transgenic larvae for selection and serial passage [32]. By microinjecting a plasmid encoding the transgene of interest, along with a plasmid encoding the *piggyBac* transposase, into the gonads of FL females and mating them with FL males, one can generate progeny in which the transgene has been randomly integrated into chromosomes at high copy number, ranging from 30–50 per genome [32]. Serial passage of selected transgenic parasites through appropriate hosts allows stable, germline-transmitting transgenic lines to be established [32].

In this study, we took advantage of this innovative system of transgenesis to generate a novel strain of transgenic *S. ratti* that expresses a known CD4+ T cell epitope. We utilized the 2W1S peptide, a variant of the Eα peptide with two tryptophans and one serine substitution at amino acid residues 3, 9, and 13, respectively [33], because there is a relatively high frequency of circulating naïve CD4+ T cells in C57BL/6 mice that recognize this epitope and a well-established MHC class II tetramer allowing for detection and purification of 2W1S-specific CD4+ T cells [34]. By expressing 2W1S peptide as a fusion protein with green fluorescent protein (GFP) and FLAG peptide in the body wall of *S. ratti*, we developed a stably transfected strain (*Hulk*) that elicits expansion of 2W1S-specfic CD4+ T cells predominantly in the lungs of infected C57BL/6 mice. Analysis of *Hulk*-expanded 2W1S-specific CD4+ T cells reveals that, contrary to bacterial 2W1S-specific CD4+ T cells, helminth 2W1S-specific CD4+ T cells represent both effector Th2 cells and suppressive Tregs. *Hulk*-elicited 2W1S-specific CD4+ T cells also preferentially release amphiregulin upon peptide restimulation, and while immunization with 2W1S peptide does not accelerate clearance of *Hulk* infection it does increase amphiregulin production in the lung. These findings suggest that CD4+ T cells specific for a body-wall helminth antigen may serve roles in immunosuppression and/or tissue repair, and that the immune response to the same peptide antigen is uniquely tuned depending on the type of pathogen encountered.

## Results

### *S. ratti* activates and expands polyclonal Th2s and Tregs, eliciting associated cytokine production

While others have demonstrated increased type 2 cytokine production and Treg expansion in gut-draining mesenteric lymph nodes (mLN) during *S. ratti* infection [35], we first sought to confirm that polyclonal CD4+ T cells in *S. ratti*-infected tissue exhibit the expected Th2 and

Treg phenotypes. *S. ratti* is a skin-penetrating soil-transmitted helminth, and once it enters the skin it travels to the lungs and naso-frontal region of the head for 1–2 days before migrating to the proximal small intestine [7]. Upon analysis of the lung at 14 days post-infection, activation markers CD11a and CD44 were both upregulated on polyclonal CD4+ T cells (Fig 1B and 1C). As expected, the frequency of GATA3+ CD4+ T cells also increased with infection (Fig 1B and 1C). Though Foxp3+ Treg expansion has been observed in the mLN of *S. ratti*-infected mice [22], we did not observe increased frequencies of polyclonal Foxp3+ and Foxp3+GATA3 + CD4+ T cells in the lungs of mice 14 days post *S. ratti*-infection relative to naïve mice (Fig 1B and 1C).

Type 2 and immunomodulatory cytokine production is enhanced in mLN and spleen during *S. ratti* infection [35], but the extent to which CD4+ T cells contribute to this cytokine production is presently unknown. To assess cytokine production by CD4+ T cells from site-draining lymph nodes, we sorted CD4+ T cells from pooled draining lymph nodes of infected mice and stimulated cells with anti-CD3/CD28 for 3 days and assessed cytokine release. As expected, CD4+ T cells robustly upregulated type 2 cytokines IL-4, IL-5 and IL-13 by day 7 post-infection (Fig 1D). This response was significantly diminished by day 14, at which point the mice cleared a majority of worms from the small intestine. In addition, we observed enhanced secretion of IL-10, as well as the EGFR ligand amphiregulin, a molecule expressed by both Tregs and Th2 cells (Fig 1D). Similar to the kinetics of type 2 cytokine production, both IL-10 and amphiregulin peaked at 7 days post-infection and had returned completely to baseline by day 14. Interestingly, there was also an increase in IFN-γ production by CD4+ T cells 7 days post-infection (S1A Fig), but this increase in type 1 cytokine production was not associated with an increase in canonical Th1 or Th17 transcription factor expression. Indeed, transcript levels of both *Tbx21* (Tbet) and *Rorc* (RORγt) were diminished in infected mice on day 7 relative to naïve mice (S1B Fig).

Since CD4+ T cells also provide help to B cells in lymph node germinal centers, we also assessed levels of polyclonal type 2-associated antibody isotypes in the serum over the course of infection. As expected, we observed an increase in total serum IgE by day 7 post-infection that was maintained at day 14 (Fig 1E). Surprisingly, we did not observe an increase in serum levels of IgG1, which is an isotype that is largely dependent on T cell help and IL-4 signaling [36]. This finding correlated with data showing an unexpected increase in IFN-γ secretion, which antagonizes IL-4 signaling and inhibits Th2-mediated inflammation (S1A Fig) [37,38]. However, IFN-γ-regulated isotypes IgG2b and IgG2c levels were not upregulated (S1C Fig), suggesting that IgE is the dominant isotype induced by *S. ratti* infection. *S. ratti* clearance was significantly impaired in RAG1-deficient mice that lack both T and B cells, as evidenced by continued fecal egg deposition and presence of intestinal worms up to 25 days post-infection (Fig 1F). Altogether, these data indicate that *S. ratti* elicits CD4+ T cells producing diverse inflammatory and immunomodulatory cytokines.

## *Hulk* is a novel strain of *S. ratti* that expresses a 2W1S-GFP-FLAG fusion protein

To create a system for *in vivo* tracking of helminth-specific CD4+ T cells, we designed a plasmid vector to encode 2W1S peptide (EAWGALANWAVDSA) as a fusion protein with GFP followed by the FLAG peptide epitope (schematic in Fig 2A). The *S. sterocoralis actin-2* promoter was used to drive high-level expression in the body wall muscle, which underlies the cuticle in *Strongyloides spp*. This construct was co-injected with a vector encoding the *piggyBac* retrotransposase into the gonads of free-living females to yield transformed, GFP-expressing progeny. After six passages through Wistar rats, GFP expression was consistently observed in

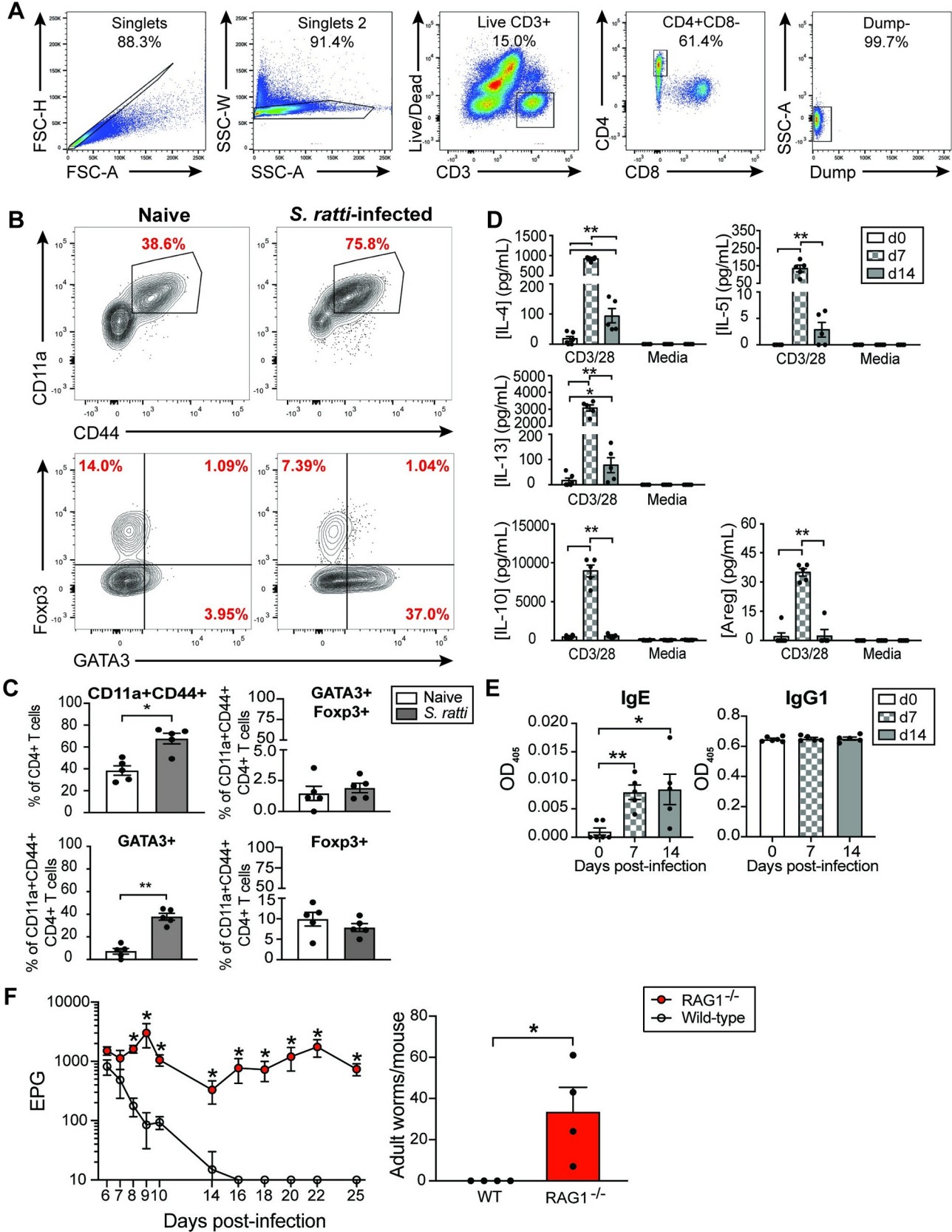

**Fig 1. Activated CD4+ T cells expand and exhibit Th2 and Treg function in during *S. ratti* infection.** (A) Gating strategy for CD4+ T cells in the lungs. Dump gate includes B220/CD45R, CD11b, and CD11c. (B) Activation status and transcription factor expression in CD4+ T cells from lungs of *S. ratti* infected mice 14 days post-initial infection (3 total infections, 1,000 iL3 per infection). (C) Frequency of activated, GATA3+, Foxp3+, and Foxp3 +GATA3+ cells in the lungs of *S. ratti*-infected and naïve mice. (D) Type 2 and regulatory cytokine production by MACS-sorted CD4+ T cells from pooled lymph nodes at 0, 7, and 14 days post-infection following 72 hours of culture in the presence or absence of anti-CD3/CD28 antibody. (E) Total serum IgE and IgG1 levels at 0, 7 and 14 days post-infection. (F) Fecal egg deposition and adult worm counts from wild-type and RAG1-/- mice infected with *S. ratti*. Significance determined by Mann-Whitney test; *p <0.05, **p < 0.01.

90–95% of *Hulk* third-stage infective larvae (iL3) (Fig 2F), and could be observed in every other free-living life stage examined (Fig 2B). To ensure that the length of the fusion protein (36 kD) was transcribed and translated in parasitic life stages, Western blotting was used to detect FLAG and/or GFP in skin-penetrating iL3, lung-residing fourth-stage larvae (L4), and gut-dwelling parasitic adult female stages of *Hulk* (Fig 2B). Consistent with microscopic analysis of the iL3 stage, *Hulk* L4 and parasitic adults both maintained GFP expression at the protein level (Fig 2D and 2E). Interestingly, microscopic analysis revealed that the frequency of GFP + parasitic adults in the small intestine (85.8% ± 2.94%) did decline relative to the frequency of GFP+ iL3 used for infection (94.3%) (Fig 2F). Further, expression of the C terminal FLAG tag was diminished in *Hulk* L4 relative to the iL3 stage (Fig 2C and 2D), and FLAG could not be detected in *Hulk* parasitic adults (Fig 2E). Importantly, neither FLAG nor GFP was detected in the parental strain in any of the life stages examined (Fig 2C–2E).

Finally, we sought to ascertain whether transgene insertion impaired the virulence of *Hulk* relative to the parental strain. We did not observe differences in worm burdens in the lung and gut on days 2 and 7 post-infection, respectively between *Hulk* and parental-infected mice (Fig 2G–2I). Lung hemorrhage, which is apparent 2 days post-infection with parental *S. ratti* and largely resolved by day 14, was comparable following infection with *Hulk* (S2 Fig). These data demonstrate that there is no appreciable difference in the infectivity of *Hulk* in comparison to parental *S. ratti*.

## *Hulk* induces 2W1S-specific CD4+ T cell expansion in the lung proportional to antigen load and GFP-specific antibody responses in serum

We next asked whether *Hulk* infection could drive adaptive immune responses against transgenically expressed antigens, specifically T cell responses against 2W1S peptide and antibodies against GFP. 2W1S-specific CD4+ T cells were found to expand in the lung parenchyma of infected mice 14 days post-infection, whereas this did not occur in mice infected with the parental strain or in naïve controls (Fig 3B). Additionally, anti-GFP IgG and IgM antibodies could be detected at higher titers in *Hulk*-infected mice than in parental-infected or naïve control mice (Fig 3C).

To determine whether the clonal burst size of *Hulk*-elicited 2W1S-specific CD4+ T cells could be increased following multiple infections, we also quantified 2W1S+CD4+ T cell expansion following three successive inoculations with 1,000 iL3 (schematic in S3A Fig). Notably, egg production kinetics over three infections was similar to that observed following a single infection, with a peak between days 6–7 post-initial infection and no subsequent peaks correspondent to subsequent infections (S3B Fig), and no adult worms were found 14 days post-infection. However, multiple live infections resulted in a 30-fold increase in the frequency and a 6.5-fold increase in the number of CD11a+2W1S+CD4+ T cells in the lung over mice given a single dose of 1,000 iL3 *Hulk* iL3 (Fig 3D). Expansion of CD11a+2W1S+CD4+ T cells also occurred to a lesser extent in the lung-draining mediastinal lymph nodes (mdLN) following one infection, and was similarly proportional to antigen load following three *Hulk* inoculations (Fig 3E). In order to corroborate this finding, we also quantified the frequency and number of

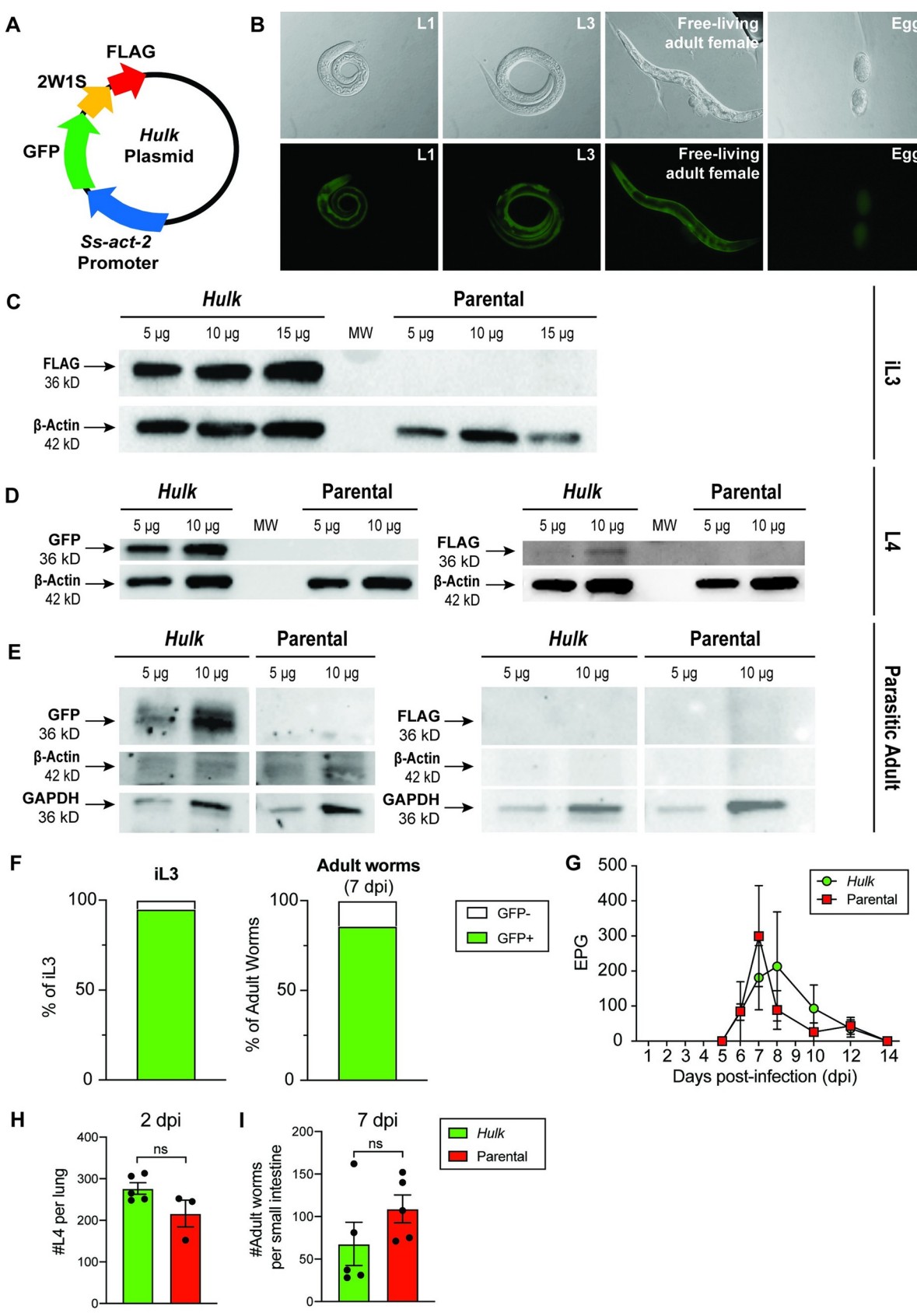

**Fig 2. *Hulk* expresses 2W1S peptide as a fusion protein with green fluorescent protein (GFP) and FLAG.** (A) Schematic depicting *Hulk* plasmid encoding green fluorescent protein (GFP), 2W1S peptide, and FLAG peptide under control of the *Strongyloides stercoralis* actin (*Ss-act-2*) promoter. (B) Brightfield DIC (top) and green fluorescent (bottom) images of the first and third larval stages, adult free-living female stage, and egg stage of *Hulk* (L1, L3, eggs: 40x magnification; Free-living adult: 20x magnification). (C-E) Western blot analysis of FLAG and/or GFP protein expression in *Hulk* and parental *S. ratti* at the indicated protein amounts in iL3 (C), L4 (D) and parasitic adult females (E). β-actin and/or GAPDH were detected as loading controls. (F) Frequency of GFP+ and GFP- parasites at the iL3 and parasitic adult female stages (7 days post-infection). (G) Fecal egg enumeration per gram of feces for C57BL/6 mice infected with1000 infective L3 (iL3) *Hulk* or the parental *S. ratti* strain (n = 4). (H-I) Enumeration of L4 (H) or parasitic adult females (I) at the indicated days post-infection. Mice were infected with 3,000 iL3 in H and with 1,000 iL3 in I. Significance tested using Mann-Whitney test.

activated and proliferating CD4+ T cells by CD11a and Ki67 expression following *in vivo* 2W1S peptide restimulation of *Hulk*-infected, parental-infected or naive mice (schematic shown in Fig 3F). Since a predominance of 2W1S-specific CD4+ T cells were found in the lung, we used irradiated larvae which arrest in the lung and delivered peptide to the lung by intratracheal injection. As expected, we observed an increase in the frequency of CD11a+Ki67+ CD4+ T cells following 2W1S peptide restimulation in *Hulk*-infected mice relative to control mice (Fig 3G–3I).

Following a molt in the lung, *S. ratti* migrates to the small intestine where it burrows into the epithelium to form sexually mature adults. Accordingly, we sought to determine whether 2W1S-specific CD4+ T cells expanded in the gut in addition to lung tissue. Interestingly, we found that irrespective of whether mice received one or three inoculations, only a negligible number of 2W1S-specific CD4+ T cells could be detected in gut-draining mLN or the Peyer's patches compared to parental-infected or naïve mice (S4 Fig). These data revealed that among the tissues evaluated, the lung parenchyma holds the predominance of the 2W1S-specific CD4+T cell response. This response pattern is also consistent with our observation that the C-terminal region of the GFP-2W1S-FLAG fusion protein is expressed in the lung-dwelling L4 but not the gut-dwelling adult stage of *Hulk*.

## 2W1S-specific CD4+ T cells exhibit distinct transcriptional profiles during *Hulk* vs. 2W1S- *Salmonella* infection

Having determined that *Hulk* infection successfully expanded 2W1S-specific CD4+ T lymphocytes, we next asked whether altering the infectious context in which the 2W1S epitope was presented would skew the phenotype and/or effector function of the CD4+ T cells. Transcriptional profiling was performed on activated 2W1S-specific CD4+ T cells isolated from mice inoculated with *Hulk*- or infected with the 2W1S-expressing vaccine strain of *Salmonella typhimurium*, LVS strain BRD509-2W1S (Δ*aroA*) (2W-*Salmonella*) (Fig 4A). *Hulk* and 2W-*Salmonella* infections drove expansion of 2W1S-specific CD4+ T cells that were transcriptionally distinct by principle component analysis, with 22.3% of variation driven by infectious context (Fig 4B). Predictably, 2W-*Salmonella* infection induced T cell-intrinsic expression of genes involved in type 1 and type 17 immune responses, including *Tbx21* (Tbet), *Rorc* (RORγt), *Il17d/f* and *Il22* (Fig 4C). However, *Hulk* infection was associated with upregulation of both canonical type 2 genes, including *Gata3*, *Il4*, *Il5*, and *Il13*, as well as immunosuppressive genes like *Il10* (Fig 4C). Though we expected these canonical type 2 and immunomodulatory genes to be highly expressed in *Hulk*-expanded 2W1S-specific CD4+ T cells, to our surprise the most significantly upregulated gene in this population was amphiregulin, an EGFR ligand known to be expressed by both Th2 cells as well as Tregs [25,39].

To further understand how two different CD4+ T cell populations with shared antigen receptor specificity were altered by infectious context, gene set enrichment analysis (GSEA) was performed using an in-house curated Th2 gene set (S1 Table) and gene ontology (GO) terms. Notably, and consistent with the differentially expressed gene list, *Hulk*-expanded

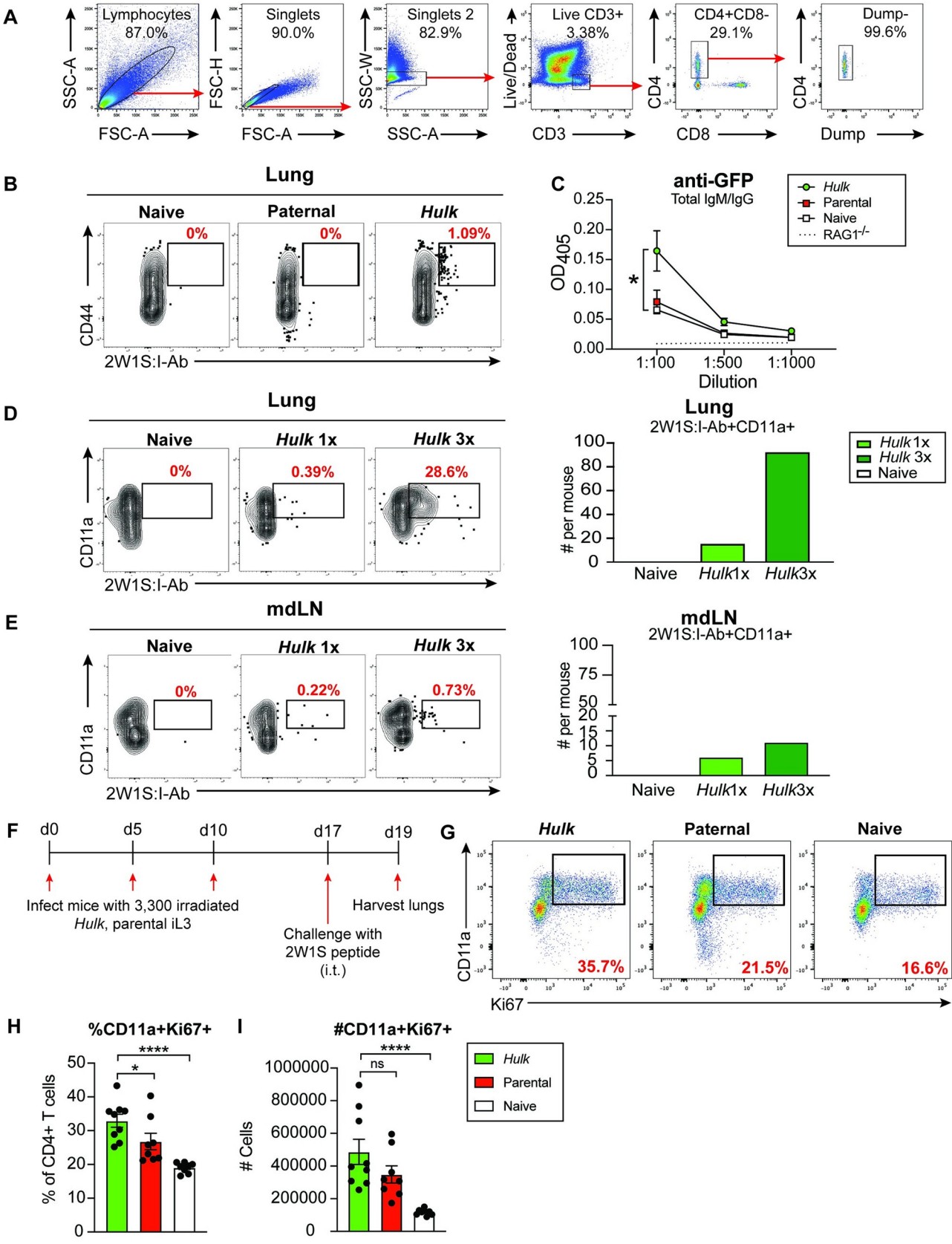

**Fig 3. *Hulk* infection induces transgene-specific adaptive immune responses.** All mice were analyzed 14 days post-initial infection, and 2W1S-specific cells were enriched using a fluorophore-specific MACS bead sorting pulldown technique prior to flow cytometry. (A) Representative gating strategy, shown in lung. Dump gate includes B220/CD45R, CD11b, and CD11c. (B) 2W1S:I-Ab+CD44+ CD4+ T cell frequency in the lungs of naive mice and mice infected once with *Hulk*, parental *S. ratti* (n = 3–4, pooled). (C) Anti-GFP IgM+IgG serum titers from naïve, parental *S. ratti*-infected and *Hulk*-infected mice, *p < 0.05. (D) 2W1S:I-Ab+CD11a+ CD4+ T cell frequency in the lungs of naive mice and mice infected once or three times (days 0, 5, 10) with *Hulk* (n = 1–3, pooled). (E) 2W1S:I-Ab+CD11a+ CD4+ T cell frequency in the lung-draining mediastinal lymph nodes (mdLN) of naive mice or mice infected once or three times with *Hulk* (n = 1–3, pooled). (F) Schematic of experimental design used in (G-I). (G) Representative flow plots showing CD11a+Ki67+ CD4+ T cells in lungs of mice infected and peptide challenged as in (F). (H-I) Frequency and number of CD11a+Ki67+ CD4+ T cells in the lungs of mice infected and peptide challenged as in (F). Significance tested using Mann-Whitney test; *p < 0.05, ****p < 0.0001.

2W1S-specific CD4+ T cells were enriched for signature Th2 genes, including transcription factors *Gata3*, *Bhlhe4* (Dec2), *JunB*, *Gfi1* and *Maf* (c-Maf), cytokines IL-4 and IL-13, and receptors for IL-33 (*Il1rl1*) and IL-4 (*Il4ra*) (Fig 4D). *Hulk*-expanded 2W1S-specific CD4+ T cells also showed multiple significantly enriched GO pathways (FDR < 0.01) (S5 Fig). Interestingly, two of these pathways were 1) Negative Regulation of IκB and NFκB signaling and 2) SMAD binding, which include genes often associated with Tregs, including *Il10*, *Tgfbr2* (TGβRII), and *Il1rl1* (IL-1Rap, a component of the IL-33 receptor) (Fig 4E). These pathways also include genes such as *Tnfaip3* (A20), which is implicated in T cell activation and activation induced cell death, and *Tob1*, a negative regulator of T cell activation, suggesting that *Hulk*-expanded 2W1S-specific CD4+ T cells may be intrinsically negatively regulated to a greater extent than those expanded during 2W-*Salmonella* infection [40,41].

GSEA revealed a number of basic cellular pathways that were upregulated in 2W1S-specific CD4+ T cells expanded by 2W-*Salmonella*, including DNA Repair and Replication (S6 Fig). Altogether, these results indicated that both *Hulk* and 2W-*Salmonella* infections drove expansion of 2W1S-specific CD4+ T cells that adopted prototypic type 2 or type 1/17 effector functions, respectively, but that *Hulk* infection specifically expanded 2W1S-specific cells with putative suppressor function. These data reveal that, whereas 2W-*Salmonella* elicited 2W1S-specific CD4+ T cells resembled type 1 and type 17 effector cells, *Hulk* elicited cells resembled type 2 effectors and immunosuppressive Tregs.

## *Hulk*-elicited 2W1S-specific CD4+ T cells produce amphiregulin in response to peptide restimulation and express both GATA3 and Foxp3

We next sought to determine the phenotype and function of 2W1S-specific CD4+ T cells during *Hulk* infection. Based on RNA sequencing, *Hulk*-elicited 2W1S-specific CD4+ T cells upregulate type 2 cytokines IL-4, IL-5, IL-13, and amphiregulin. In order to validate that this was also true at the protein level, we first performed *in vitro* 2W1S peptide restimulation of cells from lung tissue of naïve mice or mice infected 3 times with live *Hulk* or parental *S. ratti*. After 72 hours in culture, amphiregulin was significantly upregulated in cells from *Hulk*-infected mice upon peptide restimulation relative to controls (Fig 5A). Notably, the level of amphiregulin production from 2W1S stimulated cells was roughly one-third of the concentration measured in supernatants from cells polyclonally stimulated with anti-CD3, suggesting that 2W1S-specific CD4+ T cells contribute robustly to overall CD4+ T cell-derived amphiregulin levels in the lung.

To further confirm this finding, we also performed *in vivo* peptide restimulation of *Hulk*-infected, parental-infected or naïve mice. In this experiment, we infected mice 3 times with irradiated larvae, which arrest in the lung, to further enhance the amount of antigen present in this tissue (schematic shown in Fig 3F). One week after the last infection, mice were administered 2W1S peptide intratracheally, and 2 days later we isolated whole lung cells and assayed their cytokine production over 48 hours *in vitro*. As observed with *in vitro* peptide

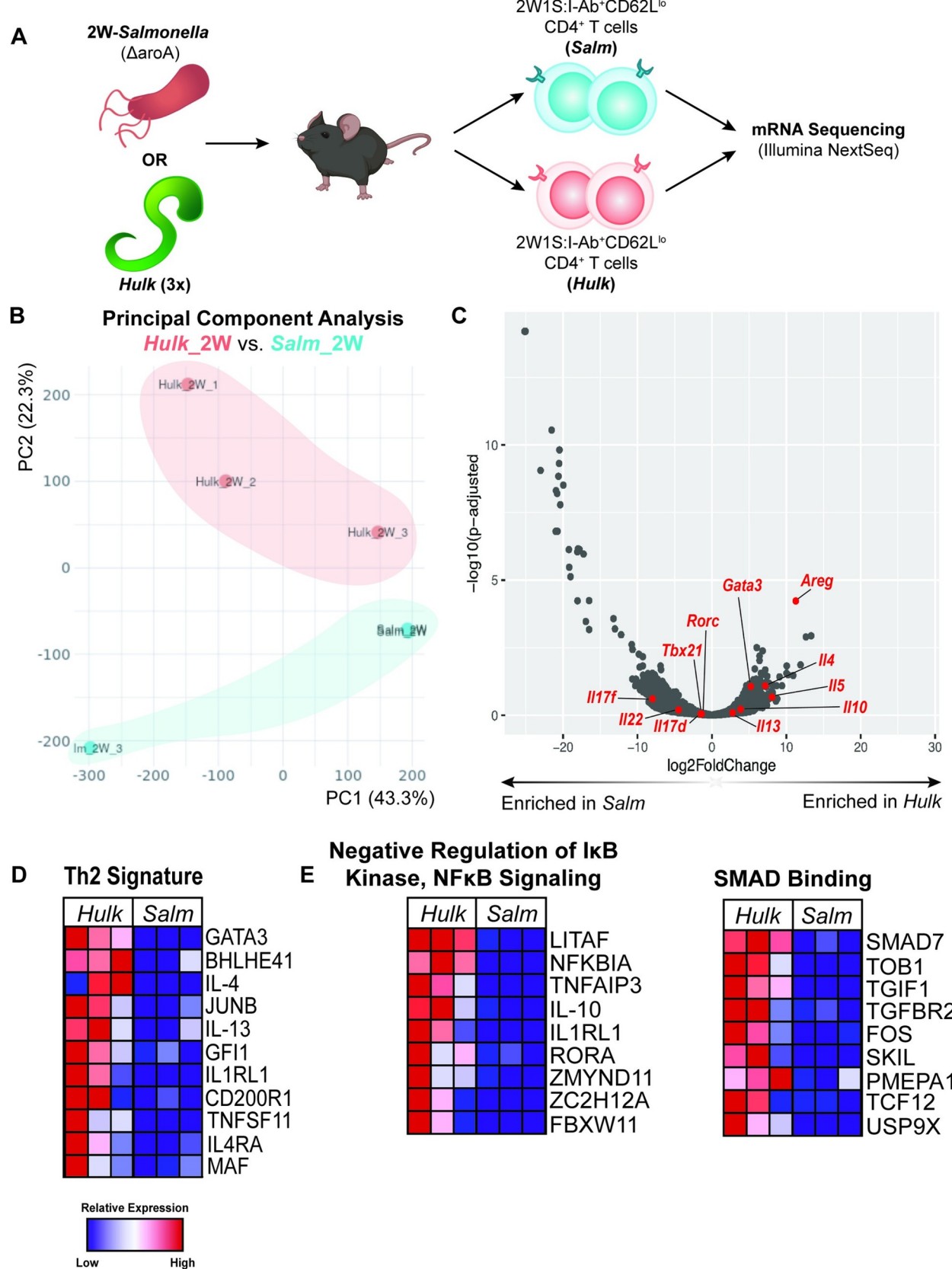

**Fig 4. *Hulk*, but not 2W-*Salmonella*, infection enriches type 2 and immunosuppressive programs in 2W1S:I-Ab+CD4+ T cells.** A) Schematic of experimental design for mRNA sequencing. 2W1S+CD4+ T cells were isolated from pooled lungs and mdLN of *Hulk*-infected mice and pooled mLN and Peyer's patches of 2W-*Salmonella*-infected mice. Schematic created with images from [BioRender.com](BioRender.com). (B) Principal component analysis comparing 2W1S-specific CD4+ T cells from *Hulk* (Hulk_2W) and 2W-*Salmonella* (Salm_2W)-infected mice. (C) Volcano plot depicting genes of interest enriched in *Hulk* or *Salm*-elicited 2W1S-specific CD4+ T cells. (D) Heatmap depicting enrichment of curated Th2 Signature gene set in *Hulk*-expanded 2W1S-specific CD4+ T cells. (E) Heatmaps depicting enrichment of Negative Regulation of IκB and NFκB Signaling and SMAD Binding gene sets (Gene Ontology) in *Hulk*-expanded 2W1S-specific CD4+ T cells.

restimulation, *in vivo* peptide restimulation of lung cells from *Hulk*-infected mice resulted in slightly enhanced amphiregulin relative to parental-infected mice and significantly enhanced production relative to naïve mice (Fig 5B). Amphiregulin levels in the bronchoalveolar lavage fluid (BALF) were also significantly elevated in *Hulk*-infected mice relative to parental controls (Fig 5C). Flow cytometric characterization of cells from BAL and lungs of these mice further suggested that the cells responding to 2W1S peptide were Th2 cells, since proliferating (Ki67+) GATA3+ CD4+ T cells were significantly enriched by frequency in *Hulk*-infected mice relative to parental-infected and naïve controls (Fig 5D–5I). Interestingly, however, type 2 cytokines IL-4, IL-5 and IL-13 were not significantly upregulated in *Hulk*-infected mice relative to controls whether stimulated *ex vivo* or *in vivo* (S7 Fig). Though surprising given the increased frequency of GATA3+ CD4+ T cells in *Hulk*-infected mice, this result is consistent with the fact that amphiregulin was more highly enriched in *Hulk* 2W1S-specific CD4+ T cells than any other type 2 cytokine by RNA sequencing.

Our RNA sequencing analysis revealed that *Hulk*-elicited 2W1S-specific CD4+ T cells upregulate GATA3 and type 2 cytokines, but also some genes associated with Tregs. Further, amphiregulin is known to be produced by both Th2 and Tregs. Accordingly, we next asked whether *Hulk*-expanded 2W1S-specific CD4+ T cells heterogeneously expressed Th2 and Treg lineage-defining transcription factors. To this end, 2W1S-specific CD4+ T cells from mice inoculated 3 times with *Hulk* were evaluated for expression of the Th2 lineage-defining transcription factor GATA3 and the Treg lineage-defining transcription factor Foxp3. Total CD4+ T cells were isolated from the lungs and lymphoid tissues of infected and naïve mice and stimulated for three days on anti-CD3/CD28-coated plates to enrich the 2W1S-specific CD4+ T cell population. We observed that a majority (60%) of 2W1S-specific CD4+ T cells from *Hulk*-infected mice were GATA3 single positive, 1.0% were Foxp3 single positive, and 1.7% were double positive for GATA3 and Foxp3 (Fig 5J and 5K). Furthermore, when 2W1S-specific CD4+ T cells were analyzed for Foxp3 expression directly *ex vivo*, up to 30% of all 2W1S-specific CD4+ T cells expressed Foxp3 (Fig 5L and 5M). In contrast, immunization with 2W1S peptide with alum as adjuvant did not expand Foxp3+ 2W1S-specific CD4+ T cells to the same extent (Fig 5L and 5M). Taken together with our transcriptional analysis, we interpret these data to indicate that *Hulk* selectively expands a heterogenous population of 2W1S-specific CD4+ T cells with the potential to function as Th2s or Tregs.

## Immunization with 2W1S peptide enhances amphiregulin-expressing CD3+ cells in the lung parenchyma following *Hulk* infection

Given that a proportion of 2W1S-specific CD4+ T cells express GATA3 and amphiregulin, a molecule necessary for clearance of the GI nematode *Trichuris muris* [39], we questioned whether these cells could promote *Hulk* clearance. However, mice immunized with 2W1S peptide and alum adjuvant had comparable worm burdens as non-immunized mice 3 and 6 days following primary *Hulk* infection (Fig 6A and 6B). This result was in spite of the increased number and frequency of activated 2W1S-specific CD4+ T cells in the lungs of immunized mice relative to non-immunized mice (S8 Fig). As expected, however, mice given a secondary

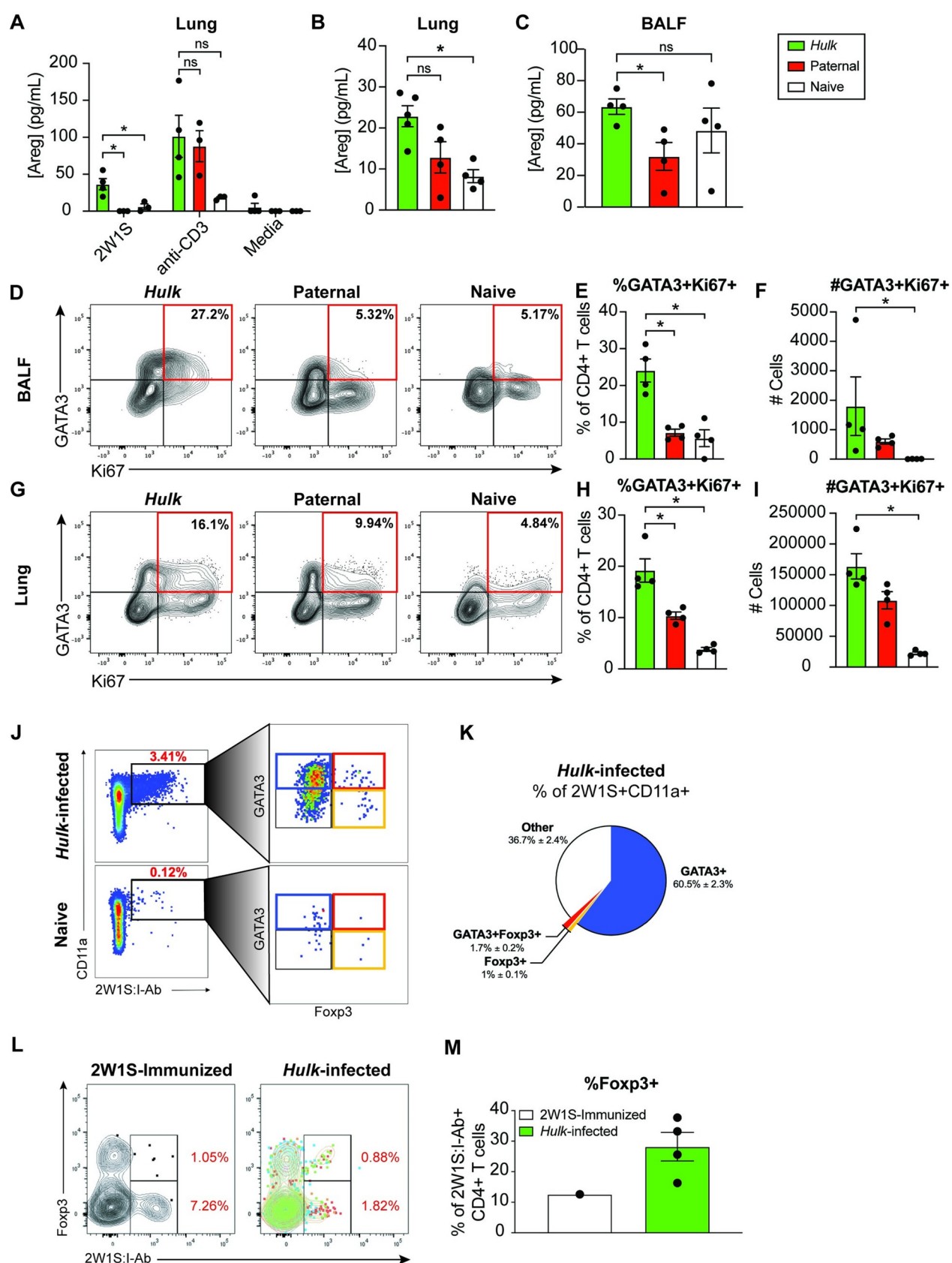

**Fig 5. CD4+ T cells that recognize 2W1S in the lung produce amphiregulin and are heterogeneously Th2 and Treg cells.** (A) Amphiregulin production by lung cells from naïve mice or mice infected 3 times with live *Hulk* or parental *S. ratti* following 72 hours restimulation with 2W1S or anti-CD3. (B) Spontaneous amphiregulin production by lung cells from naïve mice or mice infected 3 times with irradiated *Hulk* or parental *S. ratti* and challenged intratracheally with 2W1S peptide after 48 hours in culture. (C) Amphiregulin levels in broncheoalveolar lavage fluid (BALF) from mice infected and challenged as is (B). (D) Representative flow plots showing GATA3+Ki67+ CD4+ T cells in BALF of mice infected and peptide challenged as in (B). Parent gates were set according to the gating strategy shown in Fig 3A. (E-F) Frequency and number of GATA3+Ki67+ CD4+ T cells in BALF of mice infected and peptide challenged as in (B). (G) Representative flow plots showing GATA3+Ki67+ CD4+ T cells in lungs of mice infected and peptide challenged as in (B). Parent gates were set according to the gating strategy shown in Fig 3A. (H-I) Frequency and number of GATA3+Ki67+ CD4+ T cells in lungs of mice infected and peptide challenged as in (B). (J) GATA3 and Foxp3 expression in CD11a+2W1S:I-Ab + CD4+ T cells from lymph nodes and lungs of naïve mice or mice infected 3 times with live *Hulk* following 72 hours of anti-CD3/CD28 stimulation. (K) Relative frequencies of GATA3+, GATA3+Foxp3+, Foxp3+ or GATA3-Foxp3- cells within 2W1S:I-Ab+ CD4+ T cells shown in (J). (L) Foxp3-expressing 2W1S:I-Ab+ CD4+ T cells following 2W1S-immunization or 3 live *Hulk* infections. (M) Frequency of Foxp3+ cells within 2W1S:I-Ab+ CD4+ T cells. Significance determined using Student's T-test with Welch's correction (A) or Mann-Whitney test (B-C, E-F, H-I); $^*$p < 0.05.

infection with *Hulk* had significantly reduced worm burdens compared to mice given a primary infection (Fig 6A and 6B), but these mice still had significantly fewer activated 2W1S-specific CD4+ T cells than peptide-immunized mice (S8 Fig).

While amphiregulin has been implicated in GI nematode clearance, it has also been shown to be integral for wound repair at mucosal sites [24,42]. Accordingly, we sought to determine whether 2W1S immunization led to increased amphiregulin in *Hulk*-infected lungs at a time-point when larvae usually cause injury as they migrate out of the lungs and into the gut (S2 Fig) [43]. Interestingly, prior immunization with 2W1S peptide was associated with elevated spontaneous release of amphiregulin by lung cells isolated 3 days post-infection following a 72 hour culture period (Fig 6C). Additionally, we observed an increased number of both CD3 + and Areg+CD3+ cells in the lung parenchyma of 2W1S-immunized mice relative to non-immunized primary and secondary *Hulk*-infected lungs (Fig 6D–6F). Though these amphiregulin-producing T cells do not correlate with enhanced *Hulk* clearance, these data may indicate that 2W1S immunization can improve lung tissue repair later following infection.

## Discussion

To our knowledge, this is the first study utilizing a transgenic GI nematode system expressing a defined T cell epitope to study helminth-specific immunity *in vivo*. Our stably transformed nematode lined called *Hulk* allowed us to identify and characterize helminth-specific CD4+ T cells. 2W1S-specific CD4+ T cells predominantly expanded in lung tissue and lung-draining lymph nodes, but not in gut-draining lymph nodes. This expansion profile was consistent with transgene expression data showing waning C-terminal FLAG peptide expression at the protein level over subsequent life stages. As expected, a portion of 2W1S-specific CD4+ T cells expressed the canonical type 2 transcription factor GATA3 and expressed the cytokine transcripts *Il4*, *Il5* and *Il13*. However, a subset of this population also expressed Foxp3 and immunosuppressive *Il10* gene transcripts. Further, 2W1S-specific CD4+ T cells highly expressed and produced amphiregulin, an EGFR ligand which is produced by both Th2s and Tregs and has been implicated in GI nematode immunity [44]. This was distinct from 2W-*Salmonella*, where the antigen-specific cells exhibited predominantly Th1 and Th17 transcription factors and lacked evidence of an immunoregulatory phenotype. Interestingly, while 2W1S peptide immunization did not accelerate *Hulk* clearance, this treatment increased total amphiregulin levels and the number of amphiregulin-expressing CD3+ cells in the lung during *Hulk* infection. Altogether, these data demonstrate that phenotypic and functional heterogeneity within CD4 + T cells bearing a similar antigen receptor can be shaped by infectious context.

Despite the novelty and technological advancement that this work represents, there are admittedly drawbacks to the *Hulk* model. At present, *S. ratti* is one of the most tractable rodent

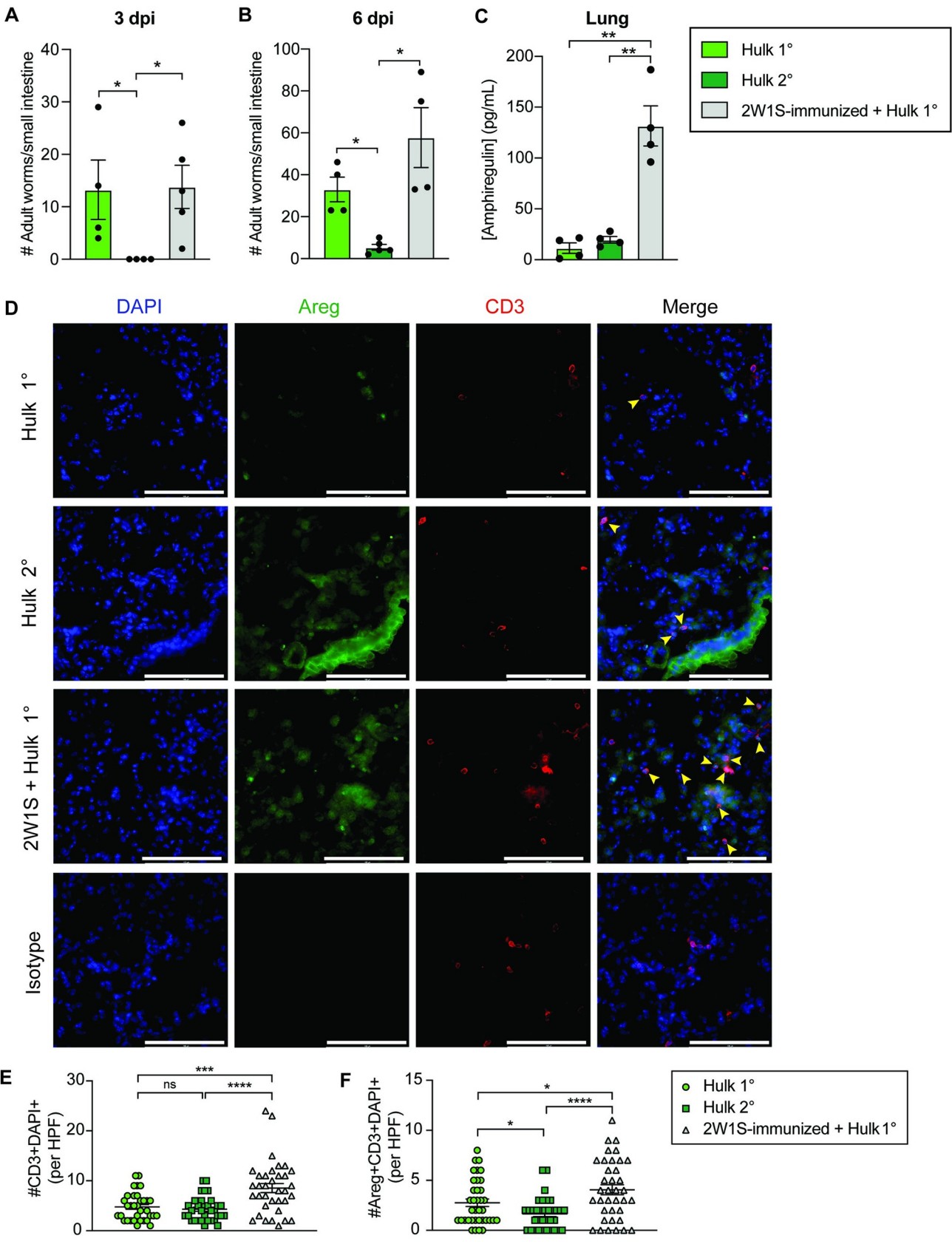

**Fig 6. 2W1S peptide immunization prior to *Hulk* infection does not confer protection, but ameliorates lung tissue damage.** Mice immunized with 2W1S peptide and alum were subsequently infected with Hulk and compared to mice given a primary or secondary Hulk infection. (A-B) Adult worm burdens in the small intestine of mice on day 3 (A) or day 6 (B) post-primary (1°) or secondary (2°) Hulk infection. (C) Spontaneous amphiregulin production in supernatants from lung cells from indicated groups isolated 3 days post-infection and cultured for 72 hours. (D) Representative images of lung tissue from 1°, 2W1S-immunized + 1° or 2° infected mice 3 days post-infection (63x). DAPI is shown in blue, CD3 is shown in red and Areg is shown in green. Yellow arrows indicate cells that were counted at DAPI+CD3+Areg+. Scale bars represent 100 um. (D) Number of CD3+DAPI+ or (E) Areg+CD3+DAPI+ cells counted per high-powered field (63x). Significance determined using (A-B) Mann-Whitney test or (C, E-F) Student's T-test with Welch's correction; *p < 0.05, **p < 0.01, ***p < 0.001, ****p < 0.0001.

parasitic nematodes in which to perform transgenic manipulation due to its clonal reproduction in host and the accessibility of its sexually reproducing free-living adult stage. However, *S. ratti* does not infect mice as effectively as other similar rodent GI nematodes, and likely does not deliver as much antigen to host tissues as other, more efficient murine pathogens (e.g. *N. brasilensis*, *H. polygyrus*). Moreover, the number of 2W1S-specific CD4+ T cells expanded by *Hulk* infection is low compared to other model antigen-expressing microbial pathogens, making it very difficult to perform adoptive transfers for functional studies. However, our work has demonstrated that these challenges can be largely overcome by a) administering multiple *Hulk* infections and b) using peptide immunization strategies to understand the functional importance of 2W1S-specific CD4+ T cells.

Whereas GFP protein is detectable in all free-living and parasitic stages of *Hulk* examined, the C-terminal FLAG tag is reduced at the protein level in the lung L4 stage and not detectable in the gut parasitic adult stage. In tandem with data showing that *Hulk* infection elicits 2W1S-specific CD4+ T cell expansion only in the lung and mediastinal lymph node and not in gut-draining lymphoid tissues, we interpret this to mean that expression of the 2W1S antigen is likely lost in the gut-dwelling adult stage. Though it is not clear why the C terminal region of the transgene is lost as *Hulk* develops within its host, there are two, non-mutually exclusive explanations: (a) the transgene, though constitutively expressed, becomes truncated or silenced at the transcript or protein level in the L4 and parasitic adult stages, and (b) the *Ss-act-2* promoter does not drive constitutive expression throughout the *S. ratti* life cycle. The first explanation is consistent with recent data showing that lentiviral-transduced *N. brasiliensis* worms can retain an mCherry transgene in their genomic DNA throughout ontogeny, but that transcription of this transgene is lost in later parasitic developmental stages of the worm [45]. Given the immunogenic nature of the 2W1S peptide, there may be selective pressure in hosts for adults to silence or edit the transgene at the transcript or protein level. Conversely, expression of this transgene in the free-living adult stages does not appear to alter survival or reproduction, and so transcription and translation of the full-length fusion protein is permissible. Further, in support of the second possibility, the free-living nematode *Caenorhabditis elegans* has 5 actin genes, and multiple actin genes have been annotated in the *S. ratti* genome [46,47]. Whether expression of different actins fluctuates throughout *S. ratti* ontogeny is not known, but it may be that each actin gene is differentially regulated at different stages of nematode development. Indeed, our data in Fig 2 show a reduction in levels of protein detected by an anti-mouse β-actin antibody in the parasitic adult stage relative to the iL3 and L4 stages, which may indicate that another actin gene is more highly expressed in this life stage. While the GFP portion of the transgene likely persists in the adult parasitic stage due to its long half-life (~26 hours), the *Ss-act-2* promoter may not drive high-level expression in the parasitic adult stage of *S. ratti*.

Nonetheless, study of helminth-specific CD4+ T cell responses in the lung still represents a tremendous utility to the field of CD4+ T cell biology and helminth immunology. The lung has previously been demonstrated as an important location for memory responses against *N. brasiliensis*, a GI nematode that mimics the infection route of *S. ratti* [48,49]. There is attrition

of larvae migrating between the lung and the intestine in primary responses, and few, if any parasites escape the lung parenchyma during secondary infection [50]. It is likely that lung-resident antigen-presenting cells have greater access to nematode body wall components from either dead or developmentally arrested larvae in the lung during primary infection, allowing for activation and infiltration of helminth body wall antigen-specific CD4+ T cells to this site. Further, the added attrition of larvae in the lungs upon secondary *N. brasiliensis* challenge is mediated by lung-resident CD4+ T cells [49]. Though they do not seem to confer protection against *Hulk* challenge, it would be interesting to investigate the degree to which 2W1S-specific CD4+ T cells are retained in the lung over time as tissue-resident memory cells post-*Hulk* infection. Since expression of the 2W1S-GFP-FLAG fusion protein is driven by an *actin-2* gene promoter that restricts expression to the body wall muscle, we are currently unable to test whether 2W1S-GFP-FLAG expression in other helminth tissue compartments or excretory-secretory (ES) products would alter the type, localization or magnitude of the immune responses generated. Indeed, polyclonal T cell responses elicited during *S. ratti* and other GI nematode infections are detected in the gut and gut-draining lymphoid tissues [22]. We are currently working on validating other *S. ratti* tissue-specific promoters that could be used to address this issue.

That a majority of 2W1S-specific CD4+ T cells in the lung express GATA3+ is consistent with the expected T cell phenotype during most helminth infections [12,51]. Th2 cells have long been thought to orchestrate worm clearance via type 2 cytokine production, but 2W1S-specific CD4+ T cells expanded by peptide immunization are not sufficient to accelerate *Hulk* expulsion. Though a protective monoclonal CD4+ T cell response against helminths would have tremendous implications for vaccination strategies, it is perhaps unsurprising that a single CD4+ T cell specificity fails to confer protection against a complex, metazoan parasite. Previous studies that compared monoclonal versus oligoclonal T cell repertoires in host immunity against *N. brasiliensis* showed neither could drive expulsion [52]. However, that work utilized TCR transgenic mice with TCRs specific for the helminth-irrelevant LCMV gp61-80 epitope or ovalbumin (OVA323-339) epitope, respectively. Thus, our study is the first to demonstrate that CD4+ T cells specific for a helminth-derived antigen do not drive worm killing or expulsion. Nevertheless, it remains a possibility that a monoclonal CD4+ T cell repertoire specific for 2W1S expressed in a difference compartment or an endogenously derived helminth epitope could promote worm clearance. Resolving the degree of T cell repertoire complexity required for protection against nematodes is a critical topic for vaccine development and is the topic of future work.

Foxp3+ Tregs have been reported to expand in *S. ratti* and other GI nematode infections [20,22], but finding that a proportion of these Tregs are helminth antigen-specific is intriguing and novel. There is evidence that both natural Tregs, defined by expression of Helios, and induced Tregs expand or differentiate in the context of helminth infections [53,54]. The latter are thought to be at least partially induced by helminth-derived secretory products that can drive Treg differentiation, many of which act through the TGF-β signaling pathway [54–56]. It would be particularly interesting to investigate whether 2W1S-specific Tregs elicited by *Hulk* infection share the same TCR sequence as 2W1S-specific Th2 effectors that arise in the same context, as this may shed light on whether Th2 vs. Treg polarization is determined by cell intrinsic (i.e. TCR) or cell extrinsic (i.e. cytokine/co-stimulatory) signals. Furthermore, the maintenance or even expansion of Tregs is relatively unique to helminth infections, as infection with intracellular microbes such as *Listeria monocytogenes* or *Toxoplasma gondii* results in disappearance of both bulk and antigen-specific Tregs during acute inflammation [29,57]. Yet the functional significance of Tregs that recognize pathogen-derived antigen, particularly in the context of helminth infection, is largely unexplored. Our data show that 2W1S-specific

Foxp3+ T cells express *Il10* transcripts, but we have not demonstrated that 2W1S-specific Foxp3+ T cells are functionally suppressive during *Hulk* infection. If helminth-specific Tregs are immunosuppressive, this could pose significant challenges to the development of lasting protective immunity following natural infection and/or vaccination. However, these cells may also be crucial for preventing tissue damage and preserving organ function, particularly in the lung [23].

We found it particularly interesting that *Hulk*-elicited 2W1S-specific CD4+ T cells upregulated amphiregulin transcript and preferentially produced amphiregulin upon peptide restimulation. Amphiregulin is necessary for immunity against the whipworm *T. muris*, and expression of amphiregulin's receptor, epidermal growth factor receptor (EGFR), is required on CD4+ T cells for clearance of *N. brasiliensis* [44]. However, data from our immunization model show that amphiregulin levels and numbers of amphiregulin-producing CD3+ T cells in the lung are not associated with reduced worm burden. Given findings showing that amphiregulin promotes epithelial cell repair and tissue remodeling in models of influenza or dextran sulfate sodium (DSS)-induced colitis, it may be that these cells instead promote enhanced tissue repair at later timepoints during *Hulk* infection [24,42]. However, whether CD4+ T cell-derived amphiregulin is necessary to promote lung repair during *S. ratti* infection remains an open avenue for investigation.

Altogether, this work generates and employs a novel transgenic *S. ratti* strain to provide the first insight into helminth-specific CD4+ T cell phenotype and potential function. This constitutes a technological advancement that offers new possibilities for the study of antigen-specific immune responses during helminth infection. In addition, *Hulk* also represents a unique tool for the study of antigen-specific Th2s and Tregs. Unlike antigen-specific cells that have been identified in other infectious contexts, 2W1S-specific CD4+ T cells elicited by *Hulk* are heterogenous and may functionally support tissue repair, though their phenotypes also suggest roles in immunosuppression and type 2 inflammation as well. We propose that *Hulk* has the potential to be used in a variety of applications, including but not limited to studies of T-B cell interactions and helminth-specific antibody production, visualization of parasite-immune interactions using fixed or live imaging, and compartmentalization or localization of CD4+ T cell responses during helminth infection.

## Materials and methods

### Ethics statement

All animal procedures were approved by the Institutional Animal Care and Use Committee (IACUC) at the University of Pennsylvania (Protocol No. 805911). Crl:WI (Wistar) rats were used for generation of the transgenic *Hulk* line. Crl:NIH-*Foxn1^{rnu}* rats and Crl:MON (Tum) gerbils were used for maintenance of both parental and transgenic *S. ratti* strains, as these hosts maintain infections for up to 6 months. All rats and gerbils were purchased from Charles River. For experimental purposes, wild-type C57BL6 mice were bred in-house or purchased from Taconic. Prior to subcutaneous infections, mice were anesthetized using isoflurane in an induction chamber. Euthanasia was performed by exposing mice to carbon dioxide ($CO_2$) for 10 minutes, after which time euthanasia was confirmed by cervical dislocation. Animals were housed under specific-pathogen free barriers in an AAALAC-accredited vivarium at the University of Pennsylvania. All IACUC protocols and routine husbandry and medical care of animals at the University of Pennsylvania were conducted in strict accordance with the Guide for the Care and Use of Laboratory Animals of the US National Institutes of Health [58].

### *S. ratti* strain generation, maintenance, and animal infection

The *Hulk* strain of *S. ratti* was derived using published methods [32]. Briefly, the GFP-2W1S-FLAG transgene contained in plasmid vector pPV691 was integrated into the parental *S. ratti* genome using the *piggyBac* transposon system, where the transposase was provided in trans by non-integrated vector pPV402. pPV691 encodes the GFP-2W1S-FLAG fusion protein under the promoter for *Ss-act-2* and utilizes the *Ss-era-1* 3' UTR, a multi-purpose terminator for transgenes in *S. stercoralis* and *S. ratti* [59,60]. Plasmid constructs are depicted in S9 Fig. Transgenic progeny were selected based on GFP-expression and passaged serially through Wistar rats until the GFP+ frequency was roughly 90–95%, which was achieved after 6 passages. Parental (ED321) and *Hulk* strains of *S. ratti* were maintained in Crl:NIH-*Foxn1^{rnu}* rats and gerbils and cultured as previously described [61]. For infections, infective third-stage larvae (iL3) were isolated via the Baermann funnel technique from charcoal cocultures grown at 22°C for 5 or more days. In cases where secondary infections were administered, mice were infected once with 1,000 live iL3 and allowed to rest for 3–6 weeks prior to challenge with 1,000 iL3 live iL3. Larvae were washed three times in phosphate-buffered saline (PBS) with 1% Penicillin-Streptomycin (P/S) to minimize carryover of fecal bacteria and enumerated under a light microscope. When indicated, iL3 were irradiated at a concentration of 16,500 iL3 per mL in PBS with 90 Gy using a Cs-137 irradiator [4]. Infective larvae were confirmed to be alive post-irradiation by motility under a microscope, and developmental arrest in-host was confirmed by absence of fecal eggs on days 6–9 post-infection. Mice were infected with indicated numbers of iL3 by subcutaneous injection of the flank skin.

### 2W-*Salmonella* culture and infection

Prior to infections, the 2W1S-expressing vaccine strain of *Salmonella Typhimurium*, LVS strain BRD509-2W1S ($\Delta aroA$), was streaked onto LB agar plates supplemented with streptomycin and incubated at 37°C overnight. LB broth supplemented with streptomycin and kanamycin was then inoculated with a single colony and incubated at 37°C in a bacterial shaker for 16–18 hours. Mice were fasted overnight prior to infection, and dosed twice orally ($5 \times 10^9$ CFU in 5% $NaCHO_3$) on days 0 and 2 and once intraperitoneally ($1 \times 10^5$ CFU in PBS) on day 2 with 2W-*Salmonella*. All mice were euthanized 7 days post-initial oral infection. Dose estimates were confirmed by serially diluting the inoculum onto LB agar supplemented with streptomycin and quantifying 24 hours later. Infection was confirmed by plating spleen homogenates from infected mice on LB agar plates supplemented with streptomycin and counting colonies 24 hours later.

### 2W1S peptide administration

For *in vivo* peptide restimulations, 100 µg 2W1S peptide (EAWGALANWAVDSA, GenScript) was administered intratracheally using a nebulizing syringe 7 days following the third of 3 infections with irradiated *Hulk* or parental *S. ratti* iL3. Mice were euthanized 2 days post-peptide challenge. For 2W1S+ CD4+ T cell identification by flow cytometry and immunization, mice were immunized with 5 or 100 µg 2W1S peptide (respectively) and 1 mg alum (Invivogen) intraperitoneally. CD4+ T cell responses were measured 5 days post-immunization, and immunized mice were challenged with Hulk 3–6 weeks following immunization.

### Tissue dissection and digestion

Bronchoalveolar lavage (BAL) was performed by flushing lungs with 0.5 mL PBS with 2 mM EDTA and 0.5% bovine serum albumin (BSA). Cells were isolated by centrifugation at 1,700

RPM for 5 minutes, and red blood cells were lysed in 1 mL ACK lysis buffer for 5 minutes at room temperature. BAL fluid (BALF) was removed after the first centrifugation for analysis by ELISA. Lungs were perfused with 10 mL cold PBS. For prefixation and cryosectioning, lungs were inflated with 0.75 mL 4% paraformaldehyde (PFA) and removed. For tissue digestion, lungs were removed and minced using dissection scissors prior to a 40 minute digest at 37°C in digestion buffer (0.4 g/L Dispase II, 0.15 g/L Liberase TM, and 0.014 g/L DNase I in serum-free DMEM). After 40 minutes, digests were extruded through a 16G needle 10 times and returned to incubate for an additional 15 minutes before being extruded again through an 18G needle 10 times. Cell suspensions were then strained over a 70 μm strainer, centrifuged at 1,700 RPM for 5 minutes at 4°C, and red blood cells were lysed in 2 mL ACK lysis buffer for 5 minutes at room temperature. Cells were centrifuged again at 1,700 RPM for 5 minutes at 4°C and resuspended in complete RPMI (cRPMI) (10% fetal bovine serum (FBS), 1% Penicillin/ Streptomycin (P/S)). Lymph nodes and Peyer's patches were dissected, homogenized over a 70 μm cell strainer (Fisher Scientific), centrifuged at 1,500 RPM for 5 minutes at 4°C and resuspended in cRPMI. Cells were enumerated using a Muse Cell Analyzer (EMDMillipore).

## Flow cytometry and Fluorescence-Associated Cell Sorting (FACS)

APC-conjugated MHCII:I-Ab EAWGALANWAVDSA (2W1S) tetramer was obtained from the NIH Tetramer Core. Tetramer-labeled 2W1S-specific CD4+ T cells were enriched and analyzed using the protocol previously described [62]. Briefly, single cell suspensions were stained for 1 hour at room temperature at a final tetramer concentration of 16.7 nM in Fc block (0.2 mg/mL goat γ globulin, 5 μg/mL 2.4G2 in PBS) and enriched over a MACS LS column using magnetic-conjugated anti-APC beads (MACS Miltenyi). Following tetramer enrichment, cells were stained with LIVE/DEAD Aqua (Invitrogen) and the following surface markers: CD3 BV650, BV711 or Pe-Cy5 (17A2 or 145-2C11, BioLegend), CD4 APC-Cy7 (GK1.5, BioLegend), CD8 PE-TexasRed (5H10, Thermo Fisher), CD44 AF700 (IM7, BioLegend), CD11a Pe-Cy7 (2D7, BD Biosciences) CD62L BV421 (MEL-14, BioLegend), B220 PerCP-Cy5.5, BV711, or FITC (RA3-6B2, BioLegend, ThermoFisher), CD11b PerCP-Cy5.5, BV711, or FITC (M1/70, BioLegend), and CD11c PerCP-Cy5.5, BV711 or FITC (N418, BioLegend, Thermo Fisher) diluted in FACS buffer (2% BSA, 0.1% NaNA₃) for 20–30 minutes. After surface stain, cells were washed with permeabilization buffer and fixed for 30 minutes using the Foxp3/Transcription Factor Staining Buffer set (ThermoFisher). After fixation, cells were washed with permeabilization buffer and stained overnight in permeabilization buffer with the following markers: Ki67 eF450 (SolA15), Foxp3 PE or Pe-Cy5.5 (FJK-16s, Thermo-Fisher), and GATA3 AF488 (TWAJ, ThermoFisher). All staining steps after tetramer enrichment were done at 4°C. All flow cytometry data were acquired on an LSR Fortessa (BD) and analyzed in FlowJo 10.6.1 (TreeStar).

For sorting, tetramer-labeled cells were enriched as for flow cytometry and stained as described above in azide-free sorting buffer (2% BSA in PBS). Roughly 1,000 live, singlet CD3 +CD4+CD8-CD62L-2W1S+ events from each biological replicate were collected into lysis buffer (SMART-seq v4 Ultra Low-Input RNA Kit, Takara Bio) in 1.5 mL Eppendorf tubes. 2W1S+CD4+ T cells were sorted from pooled lungs and mdLN of *Hulk*-infected mice, and from pooled mLN and Peyer's patches of 2W-*Salmonella*-infected mice. Cells were frozen until ready to proceed with cDNA synthesis. Sorting was performed on an Aria II (BD).

## Tissue pre-fixation and cryosectioning

Following perfusion and inflation with 4% PFA, lungs were removed with the trachea attached and incubated in 4% PFA for 1 hour at room temperature. Lungs were then washed once with

PBS and incubated in PBS for up to 4 hours at room temperature prior to being incubated overnight at 4˚C in 30% sucrose with 0.02% sodium azide. The next day, lungs were placed in 50% optimal cutting temperature (OCT; Scigen) medium with 15% sucrose for 2 hours at room temperature and then embedded in 100% OCT on dry ice.

OCT cryoblocks were sectioned at a thickness of 7μm on a Leica cryostat. Sections were placed onto poly-L-lysine-coated, positively-charged slides (Fisher Scientific) and kept frozen until immunostaining.

## Immunofluorescence

Slides were thawed at room temperature for 20 minutes and then washed 3 times in 1X tris-buffered saline (TBS) for 5 minutes each wash. Slides were then blocked and permeabilized for 1 hour at room temperature in block/perm buffer (3% BSA, 3% normal horse serum (Jackson Immunoresearch), 0.3% mouse-on-mouse blocking reagent (R&D Systems), and 0.3% TritonX-100 (Millipore-Sigma) in 1X TBS). Primary antibodies were incubated overnight at 4˚C in block/perm buffer, and include: anti-CD3 (Clone 145-2C11, eBioscience, 1:250) and anti-Amphiregulin (polyclonal, Bioss Antibodies, 1:200). The next day, slides were washed 3 times in 1X TBS for 5 minutes per wash and then incubated with the appropriate secondary antibodies for 2 hours in block/perm buffer. These antibodies include Cy-3 anti-Armenian hamster, Cy-3 anti-rat, and AF488 anti-rabbit (Jackson Immunoresearch, 1:1000). Slides were then washed as before and stained for 30 minutes with DAPI (1 μg/mL, ThermoFisher). After DAPI staining, slides were washed again as before and mounted with ProLong Gold antifade reagent (Invitrogen) and a covered with coverslips (Fisher Scientific). All slides were imaged on a DMi8 microscope (Leica), and analyzed using LAS X software (Leica) and ImageJ (NIH).

## RNA sequencing

cDNA was synthesized directly from cell lysates using the SMART-seq v4 Ultra Low-Input RNA Kit (Takara Bio). Libraries were prepared using NexteraXT library prep (Illumina) kits. The samples were loaded onto a platform for single read 75 cycles at the depth of 33 million reads per sample. Sequencing reads were trimmed for quality (phred score < 33) and to remove adapter sequences using the Trimmomatic software [63]. High quality reads were aligned to the GRCm38.p6/mm10 reference genome using the STAR aligner [64], and quantified to genes with HTSeq-count [65]. Differentially expressed genes were obtained using the DESeq2 tool in the R statistical software [66] and gene set enrichment analysis was assessed using the GSEA R tool [67,68].

## RNA isolation, cDNA synthesis and quantitative real-time (qRT)-PCR

RNA was isolated from pelleted MACS-sorted CD4+ T cells using the NucleoSpin RNA Plus kit according to the manufacturer's instructions (Macherey-Nagel). cDNA was prepared by combining RNA, random primers, and dNTPs and heating for 5 minutes at 65˚C for 5 minutes, Samples were then chilled briefly, mixed with 5x RT buffer (Invitrogen) and incubated at 25˚C for 2 minutes. Maxima H Minus reverse transcriptase (Invitrogen) was then added at a 1:1 ratio with molecular-biology grade water, and cDNA was synthesized by incubating at 25˚C for 10 minutes, 50˚C for 50 minutes, and terminating at 85˚C for 15 minutes. qRT-PCR was run and analyzed on a CFX96 platform (BioRad). Primers: *Gapdh* FWD 5'– TGTGTCCGTCGTGGATCTGA– 3', REV 5'–CCTGCTTCACCACCTTCTTGA– 3', *Rorc* (RORγt) FWD 5'–AGGAGCAATGGA AGTCGTCC– 3', REV 5'–CCGTGTAGAGGG-CAATCTCA– 3', *Tbx21* (Tbet) FWD 5'–CCAAGTTCAACC AGCACCAG– 3', REV 5'—GCCTTCTGCCTTTCCACACT– 3'.

## CD4+ T cell culture

For polyclonal CD4+ T cell stimulation, single cell suspensions from pooled infection site-draining secondary lymphoid organs (mediastinal, cervical, inguinal, and mesenteric) or lungs were sorted using the L3T4 Positive CD4+ T Cell Selection kit and LS Columns from MACS Miltenyi according to the manufacturer's instructions. After sorting, cells were washed once in MACS buffer, counted using a Muse Cell Analyzer (Millipore Sigma), and plated at a density of $1 \times 10^6$ cells per well of a 96-well round-bottom plate. Plates were either uncoated or coated with 1 µg/mL each of anti-CD3 (145-2C11, BioLegend) and anti-CD28 (37.51, BioLegend). Cells were cultured in cRPMI for 72 hours at 37˚C at 5% $CO_2$.

For whole lung cell stimulation with 2W1S peptide or anti-CD3, single cell suspensions from lungs were prepared and plated at a density of $2 \times 10^5$ cells per well in a 96 well round-bottom plate. For anti-CD3 stimulation, plates were pre-coated with 1 µg/mL anti-CD3 (145-2C11, BioLegend) and cells were resuspended in CD4+ T cell media (cRPMI + 1 mM Sodium Pyruvate, 1 mM L-Glutamine, 50 µM MEM non-essential amino acids (NEAA), 5 mM HEPES, and 50 µM β-mercaptoethanol, 10 ng/mL mouse recombinant IL-2). For peptide stimulation, cells were resuspended in CD4+ T cell media containing 100 µg/mL 2W1S peptide and 1 µg/mL anti-CD28 (37.51, BioLegend).

## Enzyme-Linked Immunosorbance Assay (ELISA)

Supernatants from cultured CD4+ T cells were harvested after 48–72 hours of culture, as described above. Analytes were probed for using the following kits: IL-4 (R&D, Invitrogen), IL-5 (Invitrogen), IL-10 (Invitrogen), IL-13 (Invitrogen), IFN-γ (Invitrogen), and amphiregulin (R&D). For serum immunoglobulin ELISAs, serum was isolated from naive mice or infected mice at the indicated timepoints and diluted 1:1000 in PBS. Plates were coated with 1 ug/mL of the appropriate anti-Ig antibody (Bethyl Laboratories, Inc.), and immunoglobulins were detected using 1 ug/mL of horseradish peroxidase (HRP)-conjugated anti-Ig antibody (Bethyl Laboratories, Inc.). For anti-GFP ELISAs, serum was isolated from naive mice or infected mice 14 days post-infection and diluted as indicated in PBS. Plates were coated with 10 ug/mL recombinant GFP (Abcam), and GFP-specific IgM and IgG were detected using 1 ug/mL of HRP-conjugated anti-IgM/IgG antibody (Jackson Immunoresearch).

## Western blotting

*Hulk* or parental *S. ratti* iL3, L4 or adult parasites were homogenized using a Dounce shaved glass homogenizer on ice in 2x RIPA buffer (Millipore-Sigma) with 1x cOmplete Mini Protease Inhibitor Cocktail (Roche). Debris was removed via centrifugation, and supernatants containing protein were quantified using a BCA assay (Invitrogen). Protein was diluted with PBS to obtain the amount indicated in Fig 2, and mixed with 4x Laemmli sample buffer (Bio-Rad) with 10% β-mercaptoethanol before being boiled for 5 minutes. Samples were then loaded onto NuPAGE 4–12% bis-tris gels (Thermo Fisher) and run in 1x NuPAGE MOPS SDS running buffer (Invitrogen) at 110V for 2 hours until the dye front reached the bottom of the gel. Protein was then transferred to a nitrocellulose membrane at 211 mAmps for 5 hours at 4˚C using NuPAGE transfer buffer (Invitrogen). Membranes were incubated in: 5% BSA in PBS + 0.1% Tween20 (Fisher Scientific) (PBS-T) for 1 hour at room temperature (RT) to block; 5% BSA in PBS-T with primary antibody for 2 hours (RT) or overnight (4˚C); and in 5% BSA in PBS-T with secondary antibody for 1–2 hours at RT. Membranes were washed 3x with PBS-T following primary and secondary antibody incubations. The following antibodies were used: mouse anti-FLAG (Clone M2, Millipore Sigma, 1:500), rabbit anti-GFP (Torrey Pines Biolabs, 1:1000), mouse anti-β-actin (Clone C4, Santa Cruz Biotechnology, 1:2000), rabbit anti-GAPDH (Bethyl Laboraties, Inc., 1:1000), HRP-linked horse anti-mouse IgG

(Cell Signaling Technology, 1:2000), and HRP-linked goat anti-rabbit IgG (Cell Signaling Technology, 1:2000). Membranes were developed using the SuperSignal West Femto Maximum Sensitivity Substrate (ThermoFisher) and exposed on a Bio-Rad ChemiDoc XRS System.

### *S. ratti* fecal egg enumeration

Fecal pellets (3–7) were collected from infected mice on the days indicated over a 10–20 minute period. Feces were weighed and then incubated for 40–60 minutes in 750 µL warm PBS at 37°C before being homogenized using a wooden stick and added to 10 mL saturated NaCl solution. Once fecal slurries were added, salt solutions were capped, shaken, and incubated at room temperature for 20 minutes. Eggs were counted by pipetting 650 µL from the meniscus of the salt float onto a McMaster slide and counting using a light microscope. Eggs per gram of feces were calculated using the following equation:

$$EPG = \frac{1}{\text{Fecal Weight}} \text{ x } \left(\frac{750}{150}\right) \text{ x No. Eggs Counted}$$

### Fourth-stage larval (L4) enumeration

Infected mice were euthanized 2 days post-infection with parental or *Hulk S. ratti*, and lungs were removed without perfusion. Lungs were minced in 7 mL PBS in 60 x 15 mm petri dishes, and incubated at 37°C for 2 hours. Following incubation, L4 were counted using a stereoscope.

### Adult *S. ratti* enumeration

At the indicated days post-infection, the entire length of the small intestine was removed from infected mice, cut longitudinally, and placed over a wire-mesh Baerman apparatus in a 250 mL beaker with enough PBS to cover the tissue. After a brief (5–10 second) agitation, tissues were incubated at 37°C for several hours to allow adult worms to migrate out of the epithelium. Adult worms that had settled to the bottom following this incubation were then counted using a stereoscope.

### Statistics

Statistical analyses were performed using Mann-Whitney tests or Student's T-test with Welch's correction in Prism7 (GraphPad). Statistical probabilities where p < 0.05 were considered significant: *p < 0.05; **p <0.01; ***p < 0.001; ****p < 0.0001.

## Supporting information

**S1 Fig. IFN-γ, but not *Tbx21*, *Rorc* or IFNγ-dependent antibodies, increase during *S. ratti* infection.** (A) IFN-γ levels in anti-CD3/CD28 (1 ug/mL) stimulated, MACS-sorted CD4+ T cells from secondary lymphoid organs of *S. ratti*-infected or naïve following 72 hours stimulation (n = 5). (B) *Tbx21* (Tbet) and *Rorc* (RORγt) expression in MACS-sorted CD4+ T cells from lungs and secondary lymphoid organs of naïve and *S. ratti*-infected mice (n = 8). (C) Total IgG2b and IgG2c antibody absorbance values in sera from naïve and *S. ratti*-infected mice (n = 5). **p<0.01.
(TIF)

**S2 Fig. Infection with parental and *Hulk S. ratti* causes gross lung pathology 2 days post-infection that is largely resolved 14 days post-infection.** Images of lungs excised from naïve or infected mice at the indicated days post-infection.
(TIF)

**S3 Fig. Experimental layout and egg deposition kinetics of three *Hulk* or parental *S. ratti* infections.** (A) Schematic of experimental layout for 3 infection model. (B) Egg production in feces (denoted as eggs per gram (EPG) over time following 3 infections with *Hulk* or parental *S. ratti*.
(TIF)

**S4 Fig. *Hulk* infection does not expand 2W1S:I-Ab+CD4+ T cells in gut-draining lymphoid tissue.** (A) 2W1S:I-Ab+CD11a+ CD4+ T cell frequency in Peyer's patches and mesenteric lymph nodes (mLN) in naive mice or mice infected once with *Hulk* or parental *S. ratti* or three times with *Hulk* (n = 3–4, pooled). (B) Number of 2W1S:I-Ab+CD11a+CD4+ in Peyer's patches and mLN in naïve, 1x *S. ratti*-infected or *Hulk*-infected, and 3x *Hulk*-infected mice.
(TIF)

**S5 Fig. Gene set enrichment analysis using GO terms for *Hulk*-expanded 2W1S-specific CD4+ T cells.** (A) Table depicting the top ten most significantly enriched gene ontology terms within 2W1S+CD4+ T cells from *Hulk*-infected mice relative to 2W1S+CD4+ T cells from 2W-*Salmonella*-infected mice. (B) Heat maps showing representative genes upregulated in 8/10 gene ontology pathways upregulated in 2W1S+CD4+ T cells from *Hulk*-infected mice in addition to those shown in Fig 4.
(TIF)

**S6 Fig. Gene set enrichment analysis using GO terms for 2W-*Salmonella*-expanded 2W1S-specific CD4+ T cells.** (A) Table depicting the top ten most significantly enriched gene ontology terms within 2W1S+CD4+ T cells from 2W-*Salmonella*-infected mice relative to 2W1S +CD4+ T cells from *Hulk*-infected mice. (B) Heat maps showing representative genes upregulated in each gene ontology pathway upregulated in 2W1S+CD4+ T cells from 2W-*Salmonella*-infected mice relative to *Hulk*-infected mice in addition to those shown in Fig 4.
(TIF)

**S7 Fig. Canonical type 2 cytokine production is unchanged in *Hulk* lungs, bronchoalveolar lavage fluid (BALF) relative to controls following 2W1S restimulation.** (A) IL-4, IL-5 and IL-13 production by lung cells from naïve mice infected 3 times with live *Hulk* or parental *S. ratti* after 72 hours stimulation with 2W1S peptide or anti-CD3. (B) Spontaneous IL-4, IL-5 and IL-13 production by lung cells from naïve mice or mice infected 3 times with irradiated *Hulk* or parental *S. ratti* and restimulated with 2W1S peptide intratracheally after 48 hours *in vitro* culture. (C) IL-13 levels in BALF of naïve mice or mice infected 3 times with irradiated *Hulk* or parental *S. ratti* and restimulated with 2W1S peptide intratracheally.
(TIF)

**S8 Fig. 2W1S-specific CD4+ T cell frequencies and numbers in the lungs of mice following secondary Hulk infection or primary Hulk infection with or without prior 2W1S immunization.** Mice immunized with 2W1S peptide and alum were subsequently infected with Hulk and compared to mice given a primary or secondary Hulk infection. (A) Concatenated flow plots showing the frequency of 2W1S+CD44+ CD4+ T cells in the lungs of each group 6 days post-challenge. Note: Lung cells were not enriched for 2W1S+ cells prior to analysis as in main paper figures. (B) Frequency and number of 2W1S:I-Ab+CD44+ CD4+ T cells in each group 6

days post-challenge. Significance was determined using a Mann-Whitney test; $^*p < 0.05$.
(TIF)

**S9 Fig. Schematic of plasmid constructs encoding GFP-2W1S-FLAG fusion protein (pPV691) and *piggyBac* transposase (pPV402).**
(TIF)

**S1 Table. Custom Th2 gene set used for gene set enrichment analysis (GSEA).**
(XLSX)

## Acknowledgments

We thank the University of Pennsylvania Flow Cytometry Core for maintenance of the cytometers used and assistance with cell sorting, the Penn Vet Center for Host-Microbial Interactions for performing RNA sequencing, the NIH Tetramer Core for providing the MHCII:I-Ab EAQGALANWAVDSA tetramer, and BioRender for the mouse image used in Fig 4.

## Author Contributions

**Conceptualization:** Bonnie Douglas, James Lok, De'Broski R. Herbert.

**Data curation:** Annabel Ferguson.

**Formal analysis:** Bonnie Douglas, Annabel Ferguson.

**Funding acquisition:** James Lok, De'Broski R. Herbert.

**Investigation:** Bonnie Douglas, Xinshe Li, Li-Yin Hung, Christopher Pastore, James Lok.

**Methodology:** Yun Wei, Xinshe Li, Jonathan R Kurtz, James B. McLachlan, Thomas J. Nolan.

**Project administration:** Bonnie Douglas, James Lok, De'Broski R. Herbert.

**Resources:** Bonnie Douglas, Xinshe Li, Christopher Pastore, Thomas J. Nolan, James Lok, De'Broski R. Herbert.

**Supervision:** James Lok, De'Broski R. Herbert.

**Validation:** Bonnie Douglas.

**Visualization:** Bonnie Douglas, James Lok, De'Broski R. Herbert.

**Writing – original draft:** Bonnie Douglas, De'Broski R. Herbert.

**Writing – review & editing:** Bonnie Douglas, Yun Wei, Li-Yin Hung, Christopher Pastore, James B. McLachlan, Thomas J. Nolan, James Lok, De'Broski R. Herbert.

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
