## [Decision Letter · Decision Letter 0]

15 May 2020

Dear Dr. Herbert,

Thank you very much for submitting your manuscript "Transgenic expression of a T cell epitope in Strongyloides ratti reveals that helminth-specific CD4+ T cells constitute both Th2 and Treg populations" for consideration at PLOS Pathogens. As with all papers reviewed by the journal, your manuscript was reviewed by members of the editorial board and by several independent reviewers. In light of the reviews (below this email), we would like to invite the resubmission of a significantly-revised version that takes into account the reviewers' comments.

Editor comments:

All reviewers are in agreement with the innovation and technical challenge of generating transgenic parasites and investigating antigen-specific T cells, and agree that these findings would be of significant impact to the filed of helminth immunologists.

However, Reviewers 1 and 2 raised important concerns that need to be addressed, notably validation of the antigen-specificity of the T cells with regards to function, and validation of transgenic expression in parasites recovered ex vivo.

Based on these concerns,

Major: three added studies are needed to

- investigate T cell antigen specificity by re-stimulation with antigen

- investigate function of the antigen-specific cells e.g. T cell transfers

- validate that worms recovered from the infected mice express the epitope

Without these added data, the conclusions of these studies may be over-interpreted, and do not sufficiently increase our understanding of mechanisms of immunity to intestinal nematode infection. These added revisions would greatly increase the significance and impact of this study.

Minor: Reviewers 2 and 3 requested a better understanding and/or discussion of the kinetics of parasite infection in lung and gut.

We cannot make any decision about publication until we have seen the revised manuscript and your response to the reviewers' comments. Your revised manuscript is also likely to be sent to reviewers for further evaluation.

Sincerely,

Meera Goh Nair

Guest Editor

PLOS Pathogens

P'ng Loke

Section Editor

PLOS Pathogens

Kasturi Haldar

Editor-in-Chief

PLOS Pathogens

orcid.org/0000-0001-5065-158X

Michael Malim

Editor-in-Chief

PLOS Pathogens

orcid.org/0000-0002-7699-2064

All reviewers are in agreement with the innovation and technical challenge of generating transgenic parasites and investigating antigen-specific T cells, and agree that these findings would be of significant impact to the filed of helminth immunologists.

However, Reviewers 1 and 2 raised important concerns that need to be addressed, notably validation of the antigen-specificity of the T cells with regards to function, and validation of transgenic expression in parasites recovered ex vivo.

Based on these concerns,

Major: three added studies are needed to

- investigate T cell antigen specificity by re-stimulation with antigen

- investigate function of the antigen-specific cells e.g. T cell transfers

- validate that worms recovered from the infected mice express the epitope

Without these added data, the conclusions of these studies may be over-interpreted, and do not sufficiently increase our understanding of mechanisms of immunity to intestinal nematode infection. These added revisions would greatly increase the significance and impact of this study.

Minor: Reviewers 2 and 3 requested a better understanding and/or discussion of the kinetics of parasite infection in lung and gut.

Reviewer's Responses to Questions

**Part I - Summary**

Reviewer #1: This is an interesting study that uses a transgenic nematode parasite to investigate CD4+T cell specific responses in vivo. Transgenesis of parasitic nematodes has been and continues to be extremely challenging technically. It is noteworthy, therefore, that this study has been able to generate such parasites and investigate immune responses to them in vivo. This is a clear strength of the study. The experiments have been carefully carried out and the analysis is sound and the data novel. However, a weakness of the work lies in that the S, ratti model in the mouse, although it has been extensively worked upon over many years, is quite variable and limited in scope. Infectivity - as defined by a productive intestinal infection - for the mouse is generally very low and variable. The, data , as presented has some deficiencies that need to be addressed. This impacts on the conclusions that can be draw from the data presented in the manuscript at the present time.

Reviewer #2: In this manuscript the authors first evaluate the T cell response that is initiated following a Strongyloides infection. During this analysis they show that infection promotes a mixed TH2 and Treg response that is characterized by the production of type 2 cytokines, IL-10 and amphiregulin. Next, the authors show that lymphocyte responses appear to play an import role in limiting parasite levels by infecting Rag K/O mice, which have elevated parasite loads. To better investigate the role of T cells in response to infection, the authors then generated a 2W1S-GFP-FLAG expressing parasite (Hulk). When mice are infected with hulk antigen-specific T cell responses are observed in the lungs, but not so much in the gut. Transcriptional analysis of these antigen-specific T cells illustrated that they have a strong TH2 cell signature mixed with a Treg signature characterized by high expression of amphiregulin. This differed substantially from T cells generated by SW1S expressing salmonella that promoted more proinflammatory T cell responses.

Reviewer #3: The paper by Douglas et al. is focused on identifying the role of CD4+ T cells in helminth infection. They begin by providing background of the importance of CD4+ T cells in controlling and clearing nematode parasites of the GI tract, and that the nature of CD4+ T cells specific for helminth antigens is unexplored. They developed a wonderful new resource to study helminth infection by producing S. ratti that expresses 2W1S, GFP, and FLAG, all on the body wall of the nematode. The production of this resource required herculean effort, which is not lost on this reviewer. They present compelling data that S. ratti infection elicits CD4+ T cells which produce diverse inflammatory and immunomodulatory cytokines. They presented nice data on the development of their transgenic nematodes, confirming that the they have a stable transgenic line. They then used this new tool to study the role of infection context on the phenotype of CD4+ T cells, demonstrating that 2W1S-expressing Salmonella infection yields a different expansion of CD4+T cells than 2W1S-expression S. ratti.

Overall the paper is well written, the experimental design and progression is logical and clearly conveyed. This study represents a significant advancement to the field and will be of interest to a broad readership. I had only a few minor suggestions. Well done!

**Part II – Major Issues: Key Experiments Required for Acceptance**

Reviewer #1: The data (Fig1) from infection of mice with WT S.ratti parasites shows that a CD4+ T cell response (and Type2) occurs in the lungs on day 14 and at day 7 in site draining pooled lymph nodes. What are these nodes? Are they a mixture of lung draining and gut draining? The data from Fig1F suggests that in RAG null mice the mean intestinal worm burden is approximately 30 worms. Although no data is presented for the WT mice, it is reasonable to assume that it would be a similar number. This is 3% of the infection dose of 1000iL3. Where are all the other parasites?

In Fig 2D epg levels are shown for both WT parasites and transgenic parasites which are similar. This is taken as evidence that the WT and transgenic parasites are similar in terms of virulence. Worm burden data was not presented which would have been informative. However, it is not clear if the parasites that received transgenic parasites were indeed transgenic. This is important as the authors say that virtually all but not all the transgenic iL3 that are prepared are indeed transgenic. As the infection level of the WT infection is so low, how can the authors be sure that the worms in the gut are transgenic and expressing the Salmonella epitope? Data showing this would seem to be critical. Also, could the authors provide data of transgenic parasite infection kinetics in Rag/null mice? This would give confidence showing that a productive intestinal infection of transgenic parasites can occur in mice and that the responses seen in the gut are not due to non transgenic parasites. This also has impact on the observation suggesting that there are almost no 2W1S- specific T cells in the gut draining lymph node (S2 Fig) yet apparently transgenic parasites in the intestine (Fig 2D):draining lymph node responses are clearly evident after a similar infection level of WT parasites are present (Fig1). So this begs the question of why there are no transgenic parasite specific T cells detected if parasites and antigen are present.

The analysis of responses in the lung tissue (and draining lymphoid tissue) would suggest that there is an anti-transgenic parasite CD4+ T cell response. Sometimes the number of cells is extremely small (less than 10 cells per mouse) and it is difficult to be convinced of significance. The authors also state that there is considerable attrition of the larvae between lung to gut. The data suggests that there is a strong lung response and it is more robust after multiple infections. Is the response more robust if the larval innoculum was increased threefold and given in one dose? Is the response due to a large number of possibly dying parasites ? Does an innoculum of dead transgenic parasites generate a strong CD4+ T cell response? The authors discuss the potential role of parasite specific CD4+ T cell lung responses in host protection, but is is unclear if this is to live parasites bearing in mind the inefficiency of the infection. Why is day 14 chosen as the date for analysis? Most of the parasites will have have gone even from the intestine at this point and much earlier from the lungs?

Reviewer #2: This manuscript addresses a topic of great interest. The ability to track antigens specific T cells following a Strongyloides infection offers substantial benefits. It allows one to test vaccine strategies, to evaluate how long the T cells persists, to test the functional qualities of those T cells in otherwise naive mice, and to evaluate how those responses evolve following during trickle infections. Unfortunately the study appears very underdeveloped in its current form. The majority of the data presented is limited to characterizing the T cell responses and provides virtually the same insight as the analysis of the polyclonal T cell assays shown in Figure 1 (a mixed TH2/Treg induction). Given that the strength of this study was to determine the importance of antigen specificity, it was a bit surprising that the T cells were stimulated with anti-CD3 and anti-CD28 rather than antigen.

The authors nicely show that antigen specific T cells increase dramatically following multiple infections, but never test how protective those T cells are if they are transferred to naive animals that are later challenged. This manuscript can be advances in many ways that offer substantial insight into the many questions that the authors have mentioned in the introduction and discussion. While the generation of the model is a great advance, it needs to be used to ask questions that reveal more insight than what has already been determine by studying bulk T cell responses.

Reviewer #3: There were no major issues that I identified. The study was well designed and the conclusions are supported by the data presented.

**Part III – Minor Issues: Editorial and Data Presentation Modifications**

Reviewer #1: None

Reviewer #2: (No Response)

Reviewer #3: In the section on the effects on Hulk on 2W1S-specific CD4+ T cell expansion, the authors comment (195-201) that the lung parenchyma holds the predominance of the 2W1S-specific CD4+ T cell response. I was glad to see that in the discussion they mentioned that it is likely that lung-resident APCs have greater access to nematode body wall components. This discussion could be expanded to include mention of the number of infecting IJs, how many make it to the lung, and how many end up in the gut. But that’s not a necessary addition.

Line 126: I’m a little uncertain as to the use of “head cavity” in describing where the nematodes go after infection. I’ve just never heard that term used to describe the trachea or esophagus, but more in reference to the cranial cavity.

Line 280: change to “is mediated by lung-resident…”

Line 326: This is the first mention of DSS colitis and the meaning of “DSS” is not explained

PLOS authors have the option to publish the peer review history of their article (what does this mean?). If published, this will include your full peer review and any attached files.

Reviewer #1: No

Reviewer #2: No

Reviewer #3: No
---

## [Decision Letter · Decision Letter 1]

23 Feb 2021

Dear Dr. Herbert,

Thank you very much for submitting your manuscript "Transgenic expression of a T cell epitope in Strongyloides ratti reveals that helminth-specific CD4+ T cells constitute both Th2 and Treg populations" for consideration at PLOS Pathogens. As with all papers reviewed by the journal, your manuscript was reviewed by members of the editorial board and by several independent reviewers. The reviewers appreciated the attention to an important topic. Based on the reviews, we are likely to accept this manuscript for publication, providing that you modify the manuscript according to the review recommendations.

On resubmission, both reviewers agree that the authors have made efforts to address their original comments but there is concern that the data provided in the response suggest significant technical difficulties in using these transgenic parasites to investigate the function of antigen-specific T cells due to their limited number and the lack of data demonstrating that they provide protection. Based on these comments:

1. Text changes are recommended to (i) remove the language "revolutionize", and (ii) add a discussion of the caveats of this model including low numbers antigen-specific T cells making functional transfer experiments prohibitive, and the caveats of the S.ratti infections in mice.

2. Per recommendation of Reviewer 2, please add Figure (i) to the manuscript as a supplemental or main figure with additional data assessing of amphiregulin expression or tissue repair in the lungs of these groups of mice.

Sincerely,

Meera Goh Nair

Associate Editor

PLOS Pathogens

P'ng Loke

Section Editor

PLOS Pathogens

Kasturi Haldar

Editor-in-Chief

PLOS Pathogens

orcid.org/0000-0001-5065-158X

Michael Malim

Editor-in-Chief

PLOS Pathogens

orcid.org/0000-0002-7699-2064

On resubmission, both reviewers agree that the authors have made efforts to address their original comments but there is concern that the data provided in the response suggest significant technical difficulties in using these transgenic parasites to investigate the function of antigen-specific T cells due to their limited number and the lack of data demonstrating that they provide protection. Based on these comments:

1. Text changes are recommended to (i) remove the language "revolutionize", and (ii) add a discussion of the caveats of this model including low numbers antigen-specific T cells making functional transfer experiments prohibitive, and the caveats of the S.ratti infections in mice.

2. Per recommendation of Reviewer 2, please add Figure (i) to the manuscript as a supplemental or main figure with additional data assessing of amphiregulin expression or tissue repair in the lungs of these groups of mice.

Reviewer Comments (if any, and for reference):

Reviewer's Responses to Questions

**Part I - Summary**

Reviewer #1: The study presents data from an initial study using a transgenic nematode parasite. This is a technological step forward and shows that this approach is doable. The authors provide some interesting data to verify the validity of the approach. The authors have clearly made attempts to address my earlier concerns and conducted extra experiments which have clarified some of the issues that I had raised and made sensible rebuttals to other points.

Reviewer #2: The authors have done some nice work in the new manuscript and showed that antigen specific activation of the T cells results in areg expression. They also attempted to vaccinate the mice with peptide and looked at worm burdens in the gut. The authors show increased expansion of the T cells but never looked at areg or tissue repair. This needs to be done and they may also want to look in the lung given the previous data shown. This paper is presenting a technical achievement that the authors say "could potentially revolutionize the field of helminth biology in regards to antigen-specific immune responses." The current version of the paper still fails to show how this new model will advance our knowledge of these T cell responses beyond their known functions.

I'm not in agreement that a single experiment of 1M T cells into a Rag animal allows you to determine that 26,000 mice would be needed to run an entirely different experiment. The idea would be that you have enriched on a highly potent population of T cells that could rapidly expand in response to seeing the antigen in the new host. This is done routinely in other models and is the reason a system like this is so helpful. Even-though the logic of coming to the 26,000 number is confusing it brings to light the concern that this model is not as useful as hoped. This statement was troubling and suggests that the model is not very useful for future experiments that could lead to very exciting results.

That authors at least need to show they can detect changes in areg when they vaccinate. Maybe that will require a couple of boosters. The data shown in Figure i should then be added to the study. With data like this added it would show that the model is going to be useful in the future.

**Part II – Major Issues: Key Experiments Required for Acceptance**

Reviewer #1: None.

Reviewer #2: (No Response)

**Part III – Minor Issues: Editorial and Data Presentation Modifications**

Reviewer #1: In the discussion - lines 336-349 the authors discuss the differences in expression of the transgene and that it is lost from the adult parasitic stages. Given the nature of the discussion it would be useful to briefly discuss this in relation to the expression of free living adult female worms , which clearly do express the transgene (Fig2B). In the last paragraph of the discussion I am not sure we are at the stage of "revolutionizing" the field of helminth biology quite yet.

Reviewer #2: (No Response)

PLOS authors have the option to publish the peer review history of their article (what does this mean?). If published, this will include your full peer review and any attached files.

Reviewer #1: No

Reviewer #2: No

Figure Files:

Data Requirements:

Reproducibility:

References:

---

## [Editor Report · Decision Letter 2]

11 Jun 2021

Dear Dr. Herbert,

We are pleased to inform you that your manuscript 'Transgenic expression of a T cell epitope in Strongyloides ratti reveals that helminth-specific CD4+ T cells constitute both Th2 and Treg populations' has been provisionally accepted for publication in PLOS Pathogens.

Best regards,

Meera Goh Nair

Associate Editor

PLOS Pathogens

P'ng Loke

Section Editor

PLOS Pathogens

Kasturi Haldar

Editor-in-Chief

PLOS Pathogens

orcid.org/0000-0001-5065-158X

Michael Malim

Editor-in-Chief

PLOS Pathogens

orcid.org/0000-0002-7699-2064

The authors have addressed the comments, edited the text, and provided supportive data as requested by the reviewers and editors.
---

## [Editor Report · Acceptance letter]

1 Jul 2021

Dear Dr. Herbert,

We are delighted to inform you that your manuscript, "Transgenic expression of a T cell epitope in Strongyloides ratti reveals that helminth-specific CD4+ T cells constitute both Th2 and Treg populations," has been formally accepted for publication in PLOS Pathogens.

Best regards,

Kasturi Haldar

Editor-in-Chief

PLOS Pathogens

orcid.org/0000-0001-5065-158X

Michael Malim

Editor-in-Chief

PLOS Pathogens

orcid.org/0000-0002-7699-2064